

# Soil profile connectivity can impact microbial substrate use, affecting how soil CO$_2$ effluxes are controlled by temperature

Frances A. Podrebarac[1,4], Sharon A. Billings[2], Kate A. Edwards[3], Jérôme Laganière[1,5], Matthew J. Norwood[1,6], Susan E. Ziegler[1]

[1]Department of Earth Sciences, Memorial University, St. John's, A1B 3X5, Canada
[2]Department of Ecology and Evolutionary Biology, Kansas Biological Survey, University of Kansas, Lawrence, 66047, USA
10    [3]Natural Resources Canada, Canadian Forest Service, Ottawa, K1A 0E4, Canada
[4]now at Genetics and Sustainable Agriculture Research, U.S. Agricultural Research Service, Mississippi State, 39762, U.S.A.
[5]now at Natural Resources Canada, Canadian Forest Service, Laurentian Forestry Centre, Quebec City, G1V 4C7, Canada
[6]now at Marine Sciences Laboratory, Pacific Northwest National Laboratory, Sequim, 98382, USA

*Correspondence to:* Susan E. Ziegler (sziegler@mun.ca)

**Abstract:** Determining controls on the temperature sensitivity of heterotrophic soil respiration remains critical to incorporating soil-climate feedbacks into climate models. Most information on soil respiratory responses to

20    temperature come from laboratory incubations of isolated soils, and typically subsamples of individual horizons. Inconsistencies between field and laboratory results may be explained by labile C or N priming supported by cross-horizon exchange - an indirect effect of quantifying microbial temperature response within intact soil profiles. Here we assess the role of soil horizon connectivity, by which we mean the degree to which horizons remain layered and associated with each another as they are *in situ*, on microbial C and N substrate use and its relationship to the temperature sensitivity of respiration. We accomplished this by exploring changes in C:N, soil organic matter composition (via amino acid composition and concentration, and nuclear magnetic resonance spectroscopy), and the δ[13]C of respiratory CO$_2$ during incubations of organic horizons collected across boreal forests in different climate regions where soil C and N composition differ. The experiments consisted of two treatments: soil incubated (1) with each organic horizon separately, and (2) as a whole organic profile, permitting cross-horizon exchange of substrates

during the incubation. The soils were incubated at 5°C and 15°C for over 430 days. Enhanced microbial use of labile C-rich, but not N-rich, substrates were responsible for enhanced, whole-horizon respiratory responses to temperature relative to individual soil horizons. This impact of a labile C priming mechanism was most emergent in soils from the warmer region, consistent with these soils' lower C bioreactivity relative to soils from the colder region. Specifically, cross-horizon exchange within whole soil profiles prompted increases in mineralization of carbohydrates and more [13]C-enriched substrates and increased soil respiratory responses to warming relative to soil horizons incubated in isolation. These findings highlight that soil horizon connectivity can impact microbial substrate use in ways that affect how soil effluxes of CO$_2$ are controlled by temperature. The degree to which this mechanism exerts itself in other soils remains unknown, but these results highlight the importance of understanding mechanisms that operate in intact soil profiles – only rarely studied – in regulating a key soil-climate feedback.





## 1 Introduction

Increased understanding of the controls on soil respiration, a globally significant flux of $CO_2$ (Bond-Lamberty and Thomson, 2010; Stocker et al., 2013), and its response to temperature is required in developing global C budgets and Earth System Models. Evidence from a global scale survey suggests that temperature sensitivity of soil respiration is largely attributed to responses occurring at the level of the whole microbial community, with the greatest temperature sensitivities occurring in high C:N ratio, C-rich soils of high-latitude boreal and arctic ecosystems (Karhu et al., 2014). Congruent with these laboratory studies, temperature sensitivity of soil respiration from field experimental warming studies indicates that the greatest enhancement occurs in high latitude soils (Carey et al., 2016). Microbial mechanisms for these high latitude soil responses as well as differences between field and laboratory studies may lie within differences in what soil horizon or collection of horizons are assessed and the consequences of this on soil and microbial community composition. For example, in boreal forest soils warming appears to enhance bacterial use of labile surface soil C sources and fungal use of deeper slower-turnover soil C pools (Ziegler et al., 2013), and lower bacterial to fungal ratios appear associated with increases in the temperature sensitivity of soil respiration (Briones et al., 2014). Legacy effects of climate, evident in semi-arid lands (Hawkes et al., 2017), also appear to impact microbial enzyme activity and its response to substrate C and N availability in boreal forest soils, though temperature sensitivity of biomass-specific $CO_2$ release does not appear to change across long timescales of exposure to warming (Min et al., 2019).

Consistent with enhanced temperature sensitivity associated with higher energy of activation ($E_a$) substrate use in purified enzyme-substrate laboratory studies (Lehmeier et al., 2013), the temperature sensitivity of soil respiration has typically been attributed to the direct effects of substrate composition (Ågren and Wetterstedt, 2007) and its $E_a$ (Bosatta and Agren, 1999; Craine et al., 2010; Fierer et al., 2005) van der Meer, 2006). For example, temperature sensitivity of soil respiration can increase with depth in association with a reduction in soil organic matter bioreactivity, suggesting increased temperature sensitivity is associated with more slow-turnover, and perhaps higher $E_a$, substrates (Conant et al., 2008; Lefevre et al., 2014; Leifeld and Fuhrer, 2005). However, these findings are not ubiquitous (Fang et al., 2005; Liski et al., 1999), nor have these laboratory findings been supported by *in situ* whole-profile investigations of respiration that reveal consistent heterotrophic respiration of relatively young soil C and elevated $Q_{10}$ of soil respiration to 100 cm (Hicks Pries et al., 2017).

Some of this variation may be attributed to the approach taken in measuring microbial temperature responses such as differences by temperature and issues using the commonly used Arrhenius based models (Alster et al., 2020; Kirschbaum, 1995; Pawar et al., 2016; Schipper et al., 2014; Sierra, 2012), as well as environmental or soil factors significantly influencing these responses leading to differences in intrinsic and apparent temperature sensitivities (Davidson and Janssens, 2006). For example, mineral association of SOM confers some protection of organic matter (Baldock and Skjemstad, 2000) and can reduce temperature sensitivity of soil respiration or eliminate its association with the bioreactivity of soil organic matter (Laganière et al., 2015; Wagai et al., 2013; Zimmermann et al., 2012) while total soil respiratory responses to temperature are affected by *in situ* tree basal area (Pennington et al. 2020). However, enhanced temperature responses of soil respiration observed within whole soil profiles suggests deeper soil profiles contribute significantly to soil respiration and its temperature response (Hicks Pries et al., 2017).



This raises questions regarding soil profile attributes, such as root and dissolved organic matter inputs or microbial substrate and nutrient exchange, that may control respiratory responses not revealed in laboratory experiments where

horizons are isolated. When conducting laboratory incubations soil samples are often homogenized from within a given soil depth removing connectivity among horizons and possible cross-horizon exchange or substrate use. Physical disruption of soil has been noted to impact soil C processes (Lin, 2012) and is likely relevant to experimental variation. This represents an important discrepancy between assessments of soil respiration (i.e. global databases and relationship with potential controlling factors such as soil moisture, vegetation) and assessment of the temperature sensitivity of soil respiration often conducted on isolated horizons of soil in the laboratory.

Soil horizon attributes impacting respiratory responses may still be important in organic horizons despite the absence of significant mineral content and its influence on organic matter availability and use. For example, temperature sensitivity of soil respiration for whole organic profiles (LFH) as found *in-situ* can differ by over 30% between samples from similar boreal forests located in different climates (Podrebarac et al., 2016). This is

despite the fact that the temperature sensitivity of soil respiration from the individual horizons (L, F and H) from those same organic profiles was lower overall and did not differ by climate (Laganière et al., 2015). It is possible that these differences in temperature response are attributed to changes in microbial access to different C or N substrates among horizons when soils were incubated as a whole organic profile versus incubations of individual horizons. Labile substrates from the less degraded L horizon may enhance the use of more complex, high $E_a$ substrates via microbial priming (Cheng et al., 2014; Finzi et al., 2015; Fontaine et al., 2007; 2011) prompted by increased availability of labile C or N substrates from more surficial horizons. Differences between SOC and SON composition between climate regions are consistent with the differences in the bioreactivity of these soils (Laganière et al., 2015) and the temperature responses of respiration of the whole soil profiles (Podrebarac et al., 2016). The colder forest soils exhibit an elevated carbohydrate content (Ziegler et al., 2017) while soil organic N content and

processing, assessed through $\delta^{15}N$ and total hydrolyzable amino acid content and composition, indicates greater availability and turnover of N within the warmer climate soils along this transect (Philben et al., 2016). This suggests that the soil horizon connectivity promoted by the whole profile enabled microbial access to substrates or enhanced substrate use that promoted the observed temperature responses not observed in the isolated soil horizons. Understanding the presence or absence of cross-horizon substrate exchange and use will help determine the mechanisms driving SOM compositional changes with temperature, as well as those governing the temperature sensitivity of soil respiration.

The prevalence and relevance of cross-horizon substrate use is unknown. Differences in microbial access to distinct substrates (e.g. labile C or N-rich compounds) may affect respiratory responses to temperature as it is clear that relevant labile inputs such as rhizosphere carbon provides a significant control on soil C and N cycling (Cheng

et al., 2014; Finzi et al., 2015). For example, labile substrates mixed into soils can enhance decomposition of extant soil organic matter and the temperature response of soil respiration (Di Lonardo et al., 2019; Wang et al., 2016; Wild et al., 2016). More labile substrates are typically found in relatively surficial soil horizons, particularly in boreal forest podzols where thick organic horizons are characterized by a surface litter horizon. The transport of these labile substrates to deeper horizons, where slower-turnover organic matter is present, occurs via mobilization of



dissolved organic matter (Kaiser and Kalbitz, 2012; Kalbitz and Kaiser, 2008). Microbes also could potentially exploit neighboring horizons' substrates via hyphae or functionally similar microbial features such as Actinobacterial mycelia (Dijkstra et al., 2013; Fontaine et al., 2011). When horizons are separated, the exchange and availability of labile C substrates across horizons is inhibited, potentially altering microbial decomposition processes and their response to temperature.

The relative availability of N can also greatly impact the strategies of microbial communities, their response to temperature and resulting rates of respiration (Billings and Ballantyne, 2013). Availability of soil N and its C:N ratio changes with soil depth, thereby representing another feature potentially responsible for differences in the microbial respiratory response to temperature between whole soil profiles and the sum of the same soils incubated in isolation. In forest soils, increased N availability can enhance substrate use by bacteria relative to fungi and Actinobacteria, and can reduce soil respiration rates (Butnor et al., 2003; Ziegler and Billings 2011l Maier and Kress, 2000). Given these observations of N-driven alterations in microbial strategies, and the relatively low bacterial to fungal ratios in boreal forest soils often associated with increased temperature sensitivity of microbial activity (Briones et al., 2014) we may expect increased N to enhance temperature sensitivity of soil respiration in boreal forest soils.

Soil connectivity may further modify these responses as more fungal dominated communities in surface horizons may access N-rich, higher $E_a$ substrates, from deeper soil horizons a mechanism found to support priming effects in some soils (Li et al., 2017). Further, use of the slower turnover substrates with low C:N ratio at depth may be coupled to the degradation of more labile, high C:N ratio substrates by the more fungal dominated community in surface horizons. This could be facilitated by N made available through microbial processing of deeper soil horizons accessed via fungal hyphae extending across soil horizons, and perhaps stimulated under more N-limiting conditions when fungi are noted to support enhanced priming effects (Dijkstra et al., 2013). Understanding how soil organic N processing relates to temperature responses of soil respiration and how the soils are incubated (i.e., as either whole organic profiles or individual isolated horizons) will enable insights into the role of cross-horizon N use in regulating the temperature sensitivity of soil respiration.

By following soil C and N use in soils from a boreal forest transect where SOC and SON composition differ both by depth and climate region, we addressed two hypotheses describing how whole soil profile connectivity affects the temperature response of soil respiration. Firstly, priming of soil respiration and C loss from slower turnover F and H horizon soils is induced by use of more labile C from the overlying L horizon and greater N availability from those deeper F and H horizons, enhancing respiratory responses to increased temperature. Secondly, the evidence for a labile C priming mechanism is most emergent within the warmer region soils given these soils' lower SOC bioreactivity relative to soils from the colder region. However, we also anticipate that if the priming mechanism is supported by cross-horizon N availability it would be most emergent within the colder region soils, where SON availability is lower relative to the warmer region. By assessing changes in C:N, soil C composition via nuclear magnetic resonance spectroscopy (NMR), soil N composition via amino acid profiling, and

$\delta^{13}$C of respiratory $CO_2$, we investigated whether the elevated $Q_{10}$ of soil respiration within the whole organic





profiles, as observed in Podrebarac et al. (2016), was associated with increased use of more labile soil C or N relative to horizons incubated individually.

## 2 Methods

### 2.1 Study Area

This study was conducted using soils from the two end-member climate regions of the Newfoundland and Labrador Boreal Ecosystem Latitudinal Transect (NL-BELT) where mean annual temperature differs by approximately 5.2°C and mean annual precipitation is 1074 and 1505 mm, for the highest latitude region (hereafter referred to as the cold region) and lowest latitude region (hereafter referred to as the warm region), respectively (Cartwright and Doyles, NL weather station climate normals between 1981-2010; Environment-Canada, 2014;

Table S1). Within these two climate regions, three mesic forest sites dominated by mature balsam fir stands (*Abies balsamea* L.) and underlain by humo-ferric podzols were established. The forest transect sites used here, established in 2011, provides a unique opportunity to determine the impact of climate history of soil organic matter cycling and the fate of *in-situ* reservoirs (Ziegler et al. 2017).

### 2.2 Soil sampling

Soil was collected in October 2011 as described in Laganière et al. (2015). Briefly, three soil sampling plots each with a diameter of 10 m were established within each site. A 20 x 20 cm intact 'cake' of the whole profile including the L, F, and H horizon was collected using a sharp knife and a trowel within each soil sampling plot. The Canadian Soil Classification of L, F, and H horizons is synonymous with $O_i$, $O_e$ and $O_a$ sub-horizons, respectively, in the U.S. Soils Classification and collectively hereafter will be referred to as the 'organic profile' or the LFH.

Although they are technically horizons of the O layer in the Canadian Soils Classification, the L, F and H will be referred to as sub-horizons from here on in order to make it easier to associate with the more commonly used U.S. Soils Classification. Mean LFH depth is 8.1 ± 0.3 cm (L 1.0 ± 0.0 cm, F 6.1 ± 0.3 cm, H 1.0 ± 0.0 cm) and 8.4 ± 0.4 cm (1.0 ± 0.0 cm, 6.3 ± 0.4 cm, 1.1 ± 0.1 cm for the L, F and H, respectively) for the cold and warm regions, respectively. On site, half of the intact organic profile was separated by hand into the L, F, and H horizons and placed in a cooler while field sampling. Samples were transported to the laboratory and stored at 5°C until analysis and experimental set-up prior to the incubation as described in Laganière et al. (2015) and Podrebarac et al. (2016). Briefly in preparation for the incubation, the L was homogenized by cutting the large soil pieces into 1 cm lengths; whereas the F and H were homogenized separately by soil sampling plot through the use of a 6mm sieve. If present, large roots (>6 mm) were removed. For the incubation, the homogenized soils were pooled by site to yield three site

composite samples per region.

### 2.3 Soil incubations




As described in Laganière et al. (2015) and Podrebarac et al. (2016), the microcosms consist of a plastic tube (5 cm diameter x 15 cm height) with an acid-washed glass wool plug and V-notches on one end to enable aeration and drainage of the soil samples. The selected soil horizon(s) are placed on the glass wool horizon inside the plastic tube and the tube placed in a 1 L mason jar with these microcosms allowed to equilibrate for 1 week at 5 °C to reduce handling effects (Robertson et al., 1999). Replicate microcosms were then incubated at 5 °C and 15 °C, the average range in temperatures across the study sites in April-August (growing season). To maintain gravimetric water holding capacity at approximately 70% for between 438 and 482 days, depending on the experiment, the soil
moisture was adjusted weekly by adding water to the top of the soil core, based on mass loss measured for each microcosm. Laganière et al. (2015) incubated the L, F, and H horizons separately to determine the soil bioreactivity and $Q_{10}$ of soil respiration specific to each separate horizon, an approach hereafter referred to as the 'isolated' experiment.  In Podrebarac et al. (2016), the organic profile was reconstructed in the same proportions by mass as found *in-situ* using the same soils used in the isolated experimental treatment; an approach hereafter referred to as 'whole' experiment. The temperature sensitivity of respiration rates were taken directly from Laganière et al. (2015) and Podrebarac et al. (2016) and reported as $Q_{10}$ simply calculated as the ratio of cumulative respiration over the course of the entire incubation at 15°C over that measured in the 5°C incubations. This simple approach was chosen because it avoids possible bias introduced when fitting data to a least-squares regression line (Sierra, 2012) enabling us to illustrate a direct comparison in the temperature sensitivity of respiration among treatments with which to place
the soil compositional results from this study.

The whole experiment was incubated in triplicate by site over the 438+ day incubation and compared with results of Laganière et al. (2015). The summation of the results of the isolated experiment using the same proportions of L, F and H horizons as in the whole experimental treatment were expected to represent the organic profile without the cross-horizon exchange that exists naturally *in-situ* in whole horizons. This summed approach is hereafter referred to as the 'predicted whole' experiment. The predicted whole values for a given soil metric, such as respiration or %C was derived using Eq. (1),

$$X_{predicted\ whole} = X_L \frac{(M_L)}{(M_{whole})} + X_F \frac{(M_F)}{(M_{whole})} + X_H \frac{(M_H)}{(M_{whole})} \qquad (1)$$

where $X_{predicted\ whole}$ represents the value of X (e.g. %C, respiration rate, C:N) for the whole experiment soil treatment as predicted from measurements of X from the isolated horizons reported in Laganière et al. (2015) or this current study. The "M" refers to the total mass of dry soil with subscripts designating the isolated sub-horizon (L, F, or H)
incubated. $M_{whole}$ represents the total soil dry mass for the incubated microcosm of the whole experiment. Using this equation we generated a predicted measurement of each soil metric (e.g. soil %N, C:N, total hydrolyzable amino acid content, ratio of alkyl-C to O-alkyl-C) for the whole profile without cross-horizon exchange with which to make direct comparisons with values observed for the actual or measured whole LFH profiles.

## 2.4 Soil respiration and the δ¹³C of respiratory CO₂

The $CO_2$ production rate and δ¹³C of respiratory $CO_2$ (δ¹³C-$CO_2$) measured at 6 and 3 time points, respectively, occurred over the course of the 438+ day incubation according to Laganière et al., (2015) and





Podrebarac et al. (2016). Briefly, each mesocosm was flushed with ambient air before being sealed with an airtight lid with a rubber septum. Prior to gas sampling a volume of $N_2$ was injected into the sealed 1 L mesocosm to avoid the generation of a vacuum upon sampling. A gas tight syringe was then used to sample the initial gas sample at the same volume as the injected $N_2$. The final gas sample was collected following the same method after 16 hours and 4 hours, respectively, for 5°C and 15°C treatments. The samples collected were injected into evacuated gas tight vials (Labco Limited, Lampeter, UK) and stored (less than 2 weeks) alongside standards until analysis for $CO_2$ concentration using an Agilent 6890A gas chromatograph with a thermal conductivity detector (Agilent Technologies, Santa Clara, CA, USA). The $CO_2$ production rate was calculated as the difference in the headspace $CO_2$ concentration between final and initial gas samples per gram of initial soil C per unit of time (mg $C$-$CO_2$ $g^{-1}$ initial C $h^{-1}$). In the case of the whole experiment, $CO_2$ production was measured on day 0, 7, 42, 96, 149, 243, and 438. The $CO_2$ production rate for the isolated experiment was measured on day 0, 7, 42, 91, 156, 245, and 482 as reported in Laganière et al. (2015). The $\delta^{13}C$-$CO_2$ measured on days 0, 96, and 438 and day 0, 91, and 482, respectively, for the whole and isolated experiments was determined on an Agilent 6890 gas chromatograph (Agilent Technologies, Santa Clara, CA, USA) using a Carboxen 1010 PLOT column (30m X 0.32mm X 15 mm; Sigma Aldrich) interfaced to a Delta V+ isotope ratio mass spectrometer (ThermoFinnegan). To determine the $\delta^{13}C$ of the $CO_2$ produced via respiration over the entire incubation a linear extrapolation through measured values over the incubation was used to obtain $\delta^{13}C$ of respired $CO_2$ on days not measured. These values were used to estimate the $\delta^{13}C$ of the total cumulative $CO_2$ over the entire incubation period using Eq. (2),

$$\delta^{13}C \text{ of total cumulative respired } CO_2 = \sum_{n=1}^{6} \left( \frac{X_n - X_{n-1}}{X_6} \right) \times Y_n \qquad (2)$$

where X is the cumulative respiration (mg $C$-$CO_2$ $g^{-1}$ initial C) measured for a given sampling time point $n$. Y is the $\delta^{13}C$ of respired $CO_2$ at each sampling time point $n$.

**2.5 Soil chemistry**

All initial and final soil samples were analyzed for %C, %N, $\delta^{13}C$, $\delta^{15}N$, total hydrolyzable amino acids (THAA), and relative differences in the proportion of main C functional groups via nuclear magnetic resonance spectroscopy (NMR). Soil samples were air-dried and ground prior to analyses. The %C, %N, $\delta^{13}C$, and $\delta^{15}N$ were analyzed with a Carlo Erba NA1500 Series II elemental analyzer (Milan, Italy) coupled to a DeltaV Plus isotope ratio mass spectrometer via a Conflo III interface (Thermo Scientific). Solid-state cross polarization magic-angle spinning (CP-MAS) experiments were performed using a Bruker AVANCE II 600 MHz with a magic-angle-spinning probe for H, C, N, and $^2H$ (MASHCCND). Samples were run at 150.96 MHz ($^{13}C$) and spun at 20kHz at 298K. Experiments run for each replicate sample (n=3) were each deconvoluted using a 19-component model within the 'DM fit' base software (Massiot et al., 2002). Chemical shift regions assigned to the following functional groups: alkyl-C (50-0 ppm), amine+methoxy-C (65-45 ppm), O-alkyl-C (90-65 ppm), di-O-alkyl-C (110-90 ppm),





aromatic-C (145-110 ppm), carbonyl-C+amide (190-165 ppm), were expressed as % of total area resolved (Preston et al., 2009; Wilson et al., 1987).

For the THAA analyses, ground samples (5-10 mg) were added to glass hydrolysis tubes followed by an addition of 1 ml of 6 M HCl. The hydrolysis tubes were sparged with $N_2$, sealed with Teflon-lined caps and heated to 110°C for 20 h. Each hydrolysis tube was opened, and an aliquot of the hydrolysate was dried under a stream of $N_2$ gas. Samples were then redissolved in 0.01 M HCl and norvaline was added as an internal standard. Amino acids were recovered from the hydrolysate using solid phase extraction and derivatized using the EZ:Faast kit for amino acid analysis (Phenomenex, USA). The derivatized samples were separated by gas chromatography using a Phenomenex ZB-AAA column (110–320°C at 30° min$^{-1}$) and quantified with a flame ionization detector using an HP6850 gas chromatograph (Agilent Technologies, Santa Clara, CA, USA). This resulting in 15 quantified amino acids (AA) including alanine, glycine, valine, leucine, isoleucine, threonine, serine, proline, aspartic acid, hydroxyproline, glutamic acid, phenylalanine, lysine, histidine, and tyrosine. Glutamine and asparagine are converted to glutamic acid and aspartic acid, respectively, during hydrolysis and are included in the measurement of these amino acids. The THAA yield was expressed as a percentage of total soil C or N based on the total of all 15 AA according to Eq. (3),

$$THAA \ (\%C \ or \ N) = \sum \left( \frac{Yield_{AA}}{(C \ or \ N)} \right) \times [Wt\% \ (C \ or \ N)_{AA}] \qquad (3)$$

where $\frac{Yield_{AA}}{C \ or \ N}$ is the C- or N- normalized yield of each AA given in mg amino acid per 100 mg C or N and $Wt\% \ (C \ or \ N)_{AA}$ is the weight %C or N in the AA.

**2.6 Statistical Analyses**

The initial soil measures (C:N, %N of THAA, mol% glycine, %alkyl-C, alkyl-C:-O-alkyl-C ratio (A:OA), and %di-O-alkyl-C, $\delta^{13}$C, $\delta^{15}$N) were analyzed using a two-way ANOVA to test the effects of climate region, horizon, and their interaction term to quantify meaningful differences in soil properties prior to these experiments. Previous work investigated the impact of climate region on the soil respiratory responses to temperature (Laganière et al., 2015; Podrebarac et al., 2016). In contrast, this study focuses on the mechanisms controlling those responses. As such, we focus on the impact of the intact nature of whole soil horizons on the temperature responses of soil respiration and its relationship to C and N substrate use. To investigate how microbial use of C and N components of the SOM pools relate to the previously reported $Q_{10}$ of soil respiration (Laganière et al., 2015; Podrebarac et al., 2016), the soil mass loss, loss of C and N, change in soil composition ($\Delta$ of C:N, $\delta^{13}$C, $\delta^{15}$N, THAA, NMR resolved C chemistry), and the $\delta^{13}$C-CO$_2$ were tracked over the incubation within the (1) isolated experiment, (2) whole experiment, and (3) predicted whole experiment based on the isolated experimental results. The difference between the initial and final soil composition metric (given as final - initial = $\Delta$) was calculated, with negative values indicating a decline in metric value over the incubation (e.g., a $\Delta$ of -10 for soil C:N indicates that C:N decreased by 10 units). Due to large standard errors associated with some of the changes in composition, a Student's T-test was used to determine if the initial and final soil values were significantly different from each other (i.e., if $\Delta$ was different from zero). Because of a significant effect of region on multiple response variables (see Results), we



performed region-specific tests to assess mechanisms responsible for respiratory responses. For the isolated experiment, the effect of incubation temperature, horizon, and their interaction was assessed using a two-way ANOVA where levels of temperature were 5 ˚C and 15 ˚C, and horizons defined as L, F, and H within each climate region. Using both the isolated and whole horizon treatments we explored the effect of incubation temperature, experiment type (i.e., whole, isolated), and their interaction using a two-way ANOVA within climate region. Here, a

significant effect of experimental type denotes some impact of cross-horizon exchange on SOM processing.

For tests in which residuals did not meet the assumptions of normality and homoscedasticity, data were log$_{10}$-transformed prior to testing (Zar 1999). The Tukey's Honestly Significance Differences test was used to determine which combination of treatments or effects were significantly different. All statistical analyses were performed using R with a significance threshold set at 0.05 (R-Core-Team, 2017, 2014).

## 3 Results

### 3.1 Initial soil organic matter chemistry and nitrogen differ between the cold and warm regions

The initial soil N content in the present study was greater in the warm relative to the cold region (p<0.0001), consistent with soil sampled in earlier studies from the same climate regions (Philben et al. 2016). The initial soil C:N ranged from 32-54. These ratios did not exhibit any region by horizon effects (p=0.1272), but were

lowest in the warm region (p<0.0001) and decreased with depth (p<0.0001; Fig. 1a). The initial %N as THAA ranged from 38-43% and was similar in soils for the two climate regions, and decreased with depth (p=0.0011; Fig. 1b) in a manner consistent with increasing soil organic N degradation with depth (Philben et al. 2016). Consistent with this feature, mol% glycine increased with depth (p<0.0001) and was elevated in the warmer region soils (p=0.0035) and to a greater extent within the H horizon (p=0.0126), consistent with less soil organic N degradation in the cold relative to the warm region (Fig. 1c; Philben et al., 2016). The initial %alkyl-C ranged from 27-33% and exhibited a region by horizon interaction (p=0.0443), revealing higher values in F relative to L, and H relative to F horizons for the warm region (Fig. 1d). Higher %alkyl-C was observed in the L and H of the warmer region (p=0.0301) that increased with depth (p=0.0137). The initial %di-O-alkyl-C, ranging from 7-9%, was elevated in the cold region soils (p<0.0001; Fig. 1e) but exhibited no other trends. The alkyl-C to O-alkyl-C ratio exhibited a

regional (p=0.0014) effect only with an elevated ratio in warm relative to cold region soils (Fig. 1f).

### 3.2 Losses of soil mass and declines in soil C and N concentrations were enhanced by increased temperature and to a greater extent in the cold region soils

Over the 438+ day incubation, mass loss, %C loss, and %N loss were greatest in the 15°C incubation and for the cold region soils (Table 1). In the isolated experiment, mass loss ranged from 16 ± 1 to 39 ± 5% and 16 ± 3 to 30 ± 2% (mean ± standard error), respectively, for the cold and warm regions. The % C loss ranged from 21 ± 3 to 82 ± 4% in the cold region soils with 82% loss at 15°C in the F sub-horizon. The warm region soils exhibited a % C loss that ranged from 24 ± 5 to 46 ± 3%. The % N loss in the cold region soils ranged from 11 ± 9 to 45 ± 5% and





was greatest in the 15°C treatment.  In the warm region % N loss ranged from 13 ± 3 to 52 ± 23% and did not

exhibit a temperature effect (p = 0.227). In the whole organic profile (whole experiment) and the predicted whole

experiment soils, %mass loss was greatest at 15°C and ranged from 16 ± 2 to 37 ± 1% and 12 ± 2 to 28 ± 2%,

respectively, for the cold and warm regions.  In the warm region, %mass loss was greater in the predicted

experiment relative to the whole profile experiment (p = 0.011). Effect of temperature and experiment was observed

for %C loss in the cold and warm regions with greatest %C losses having occurred at 15 °C and in the predicted

whole experiment. The %N loss in these whole profile experiments ranged from 11 ± 4 to 38 ± 5% in the cold

climate soils where greater loss was observed at 15 °C yet no experimental effect was observed (p = 0.119).  The

warm climate soils exhibited a similar range in % N loss from 13 ± 3 to 40 ± 13% with no temperature effect (p =

0.195) or experimental effect (p = 0.064).

**Table 1.**  The mean (standard error) of % mass loss, % C loss and % N loss within each experiment where individual horizons were incubated in isolation from each other (isolated experiment), then calculated as a whole profile values based upon those isolated horizon results (predicted), and given also as the actual measured incubation results for whole organic profiles (measured). The effect of temperature (T), horizon (H), and interaction term (T x H) are given for the isolated experimental results (Top). The effect of T, experiment (E), and interaction term (T x E) are given for the tests conducted across both the predicted and measured whole profile experimental treatments (Bottom).  Significance (α = 0.05) is denoted in bold.

| | | % Mass loss | | | | % C loss | | | | % N loss | | | |
|---|---|---|---|---|---|---|---|---|---|---|---|---|---|
| | | Cold region | | Warm region | | Cold region | | Warm region | | Cold region | | Warm region | |
| Experiment | Horizon(s) | 5°C | 15°C | 5°C | 15°C | 5°C | 15°C | 5°C | 15°C | 5°C | 15°C | 5°C | 15°C |
| Isolated | L | 22.62 | 36.18 | 18.27 | 27.11 | 34.22 | 65.80 | 26.93 | 42.93 | 12.76 | 23.36 | 12.56 | 9.16 |
| | | (3.46) | (4.68) | (1.63) | (3.52) | (6.90) | (10.13) | (4.09) | (4.54) | (9.57) | (14.75) | (3.31) | (4.92) |
| | F | 23.54 | 39.03 | 21.45 | 29.73 | 34.53 | 81.52 | 31.21 | 46.15 | 22.86 | 45.37 | 23.74 | 51.89 |
| | | (1.55) | (4.49) | (2.56) | (2.36) | (2.74) | (4.20) | (1.24) | (2.79) | (3.68) | (4.52) | (9.46) | (22.83) |
| | H | 15.81 | 27.91 | 15.75 | 22.12 | 21.05 | 56.70 | 24.10 | 42.92 | 11.43 | 27.25 | 21.95 | 31.86 |
| | | (1.13) | (2.02) | (2.48) | (1.77) | (3.23) | (12.17) | (4.66) | (3.23) | (8.49) | (5.22) | (7.46) | (6.44) |
| | | | | | | | | | | | | | |
| | Effects | F | p | F | p | F | P | F | p | F | p | F | p |
| | T | 27.43 | **0.0002** | 15.16 | **0.0021** | 39.09 | **<0.0001** | 24.42 | **0.0003** | 5.41 | **0.0383** | 1.63 | 0.2265 |
| | H | 4.84 | **0.0288** | 3.67 | 0.0572 | 3.33 | 0.0709 | 0.85 | 0.4535 | 2.16 | 0.1582 | 2.98 | 0.0887 |
| | T x H | 0.14 | 0.8709 | 0.14 | 0.8723 | 0.57 | 0.5783 | 0.12 | 0.8886 | 0.24 | 0.7891 | 1.02 | 0.3904 |
| | | | | | | | | | | | | | |
| Whole Profile | predicted | 21.97 | 36.49 | 19.84 | 27.87 | 32.02 | 74.15 | 29.14 | 44.98 | 18.94 | 38.07 | 21.38 | 40.47 |
| | | (0.20) | (1.76) | (2.08) | (2.26) | (0.79) | (0.63) | (0.89) | (2.33) | (3.89) | (4.87) | (6.72) | (13.37) |
| | measured | 15.46 | 36.82 | 12.32 | 21.81 | 21.07 | 67.62 | 21.31 | 38.04 | 10.61 | 31.70 | 13.14 | 15.78 |
| | | (1.34) | (2.24) | (2.44) | (1.18) | (1.29) | (4.73) | (3.35) | (4.19) | (3.68) | (4.30) | (3.30) | (0.98) |
| | | | | | | | | | | | | | |
| | Effects | F | p | F | p | F | P | F | p | F | p | F | p |
| | T | 129.27 | **<0.0001** | 18.30 | **0.0027** | 313.78 | **<0.0001** | 30.32 | **0.0006** | 22.81 | **0.0014** | 2.00 | 0.1946 |
| | E | 3.83 | 0.0860 | 10.99 | **0.0106** | 12.20 | **0.0082** | 6.23 | **0.0372** | 3.05 | 0.1189 | 4.60 | 0.0643 |
| | T x E | 4.70 | 0.0621 | 0.13 | 0.7312 | 0.78 | 0.4026 | 0.02 | 0.8829 | 0.05 | 0.8213 | 1.15 | 0.3151 |



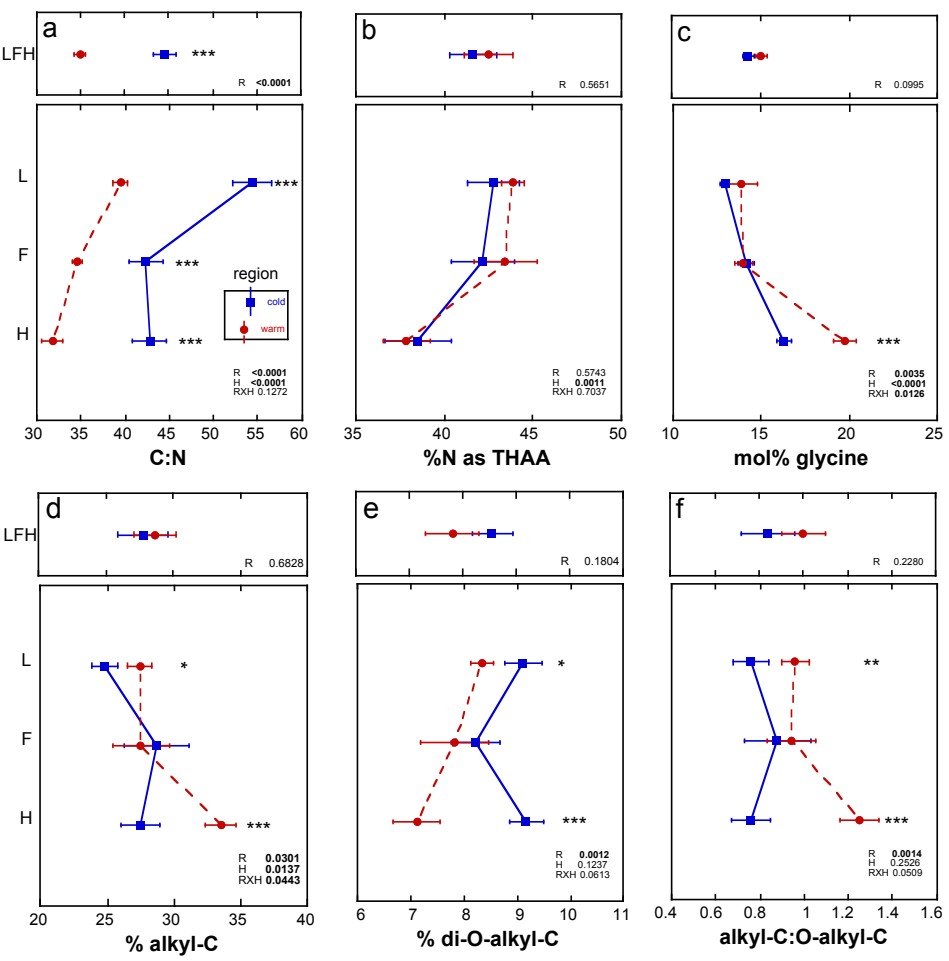

**Figure 1.** Initial mean of the three sites ± standard error of soil metrics for the whole organic profile (LFH; upper panel) and individual horizons (L, F, H) from the isolated experiment (lower panel). Within the organic profile, an analysis of variance was utilized for the effect of region (R). Significance is denoted by asterisks: $p \leq 0.0001$ ***, $p \leq 0.001$ **, $p \leq 0.05$ *, and $p > 0.05$ n.s.. The effect of R within horizon is denoted with the asterisks.

### 3.3 The $\delta^{13}C$ of cumulative respired $CO_2$ increased in whole-profile incubations, and with incubation temperature

The $\delta^{13}C$ of respired $CO_2$ increased throughout the incubation period. However, an exception to this includes all the soil horizons from the cold region incubated within the isolated experiment and, as a



result, the predicted whole experimental treatment (Fig. 2). The $\delta^{13}C$ of respired $CO_2$ over the course of
the incubation ranged from -29 to -24‰, with more $^{13}C$-enriched respired $CO_2$ released from the deeper
soil horizons and soils exposed to the higher incubation temperature. Overall the effect of temperature
($p=0.0024$), experiment ($p<0.0001$) and their interaction ($p<0.0001$) was observed in the cold region soils
while only the effect of experiment ($p = 0.0010$) was observed in the warm region soils. Regardless of
region or temperature, the whole experiment exhibited more $^{13}C$-enriched respired $CO_2$ akin to the impact
of increased temperature across all soils and experiments. For example, the whole experiment incubations
at both temperatures exhibited a 3-4‰ increase in $\delta^{13}C$-$CO_2$ relative to both the initial and the isolated
experiment at 5˚C (Fig. 2b). In contrast, the cold region soils incubated as individual L, F and H horizons
exhibited a temperature difference with a ~3‰ increase in $\delta^{13}C$-$CO_2$ in the 15˚C relative to both

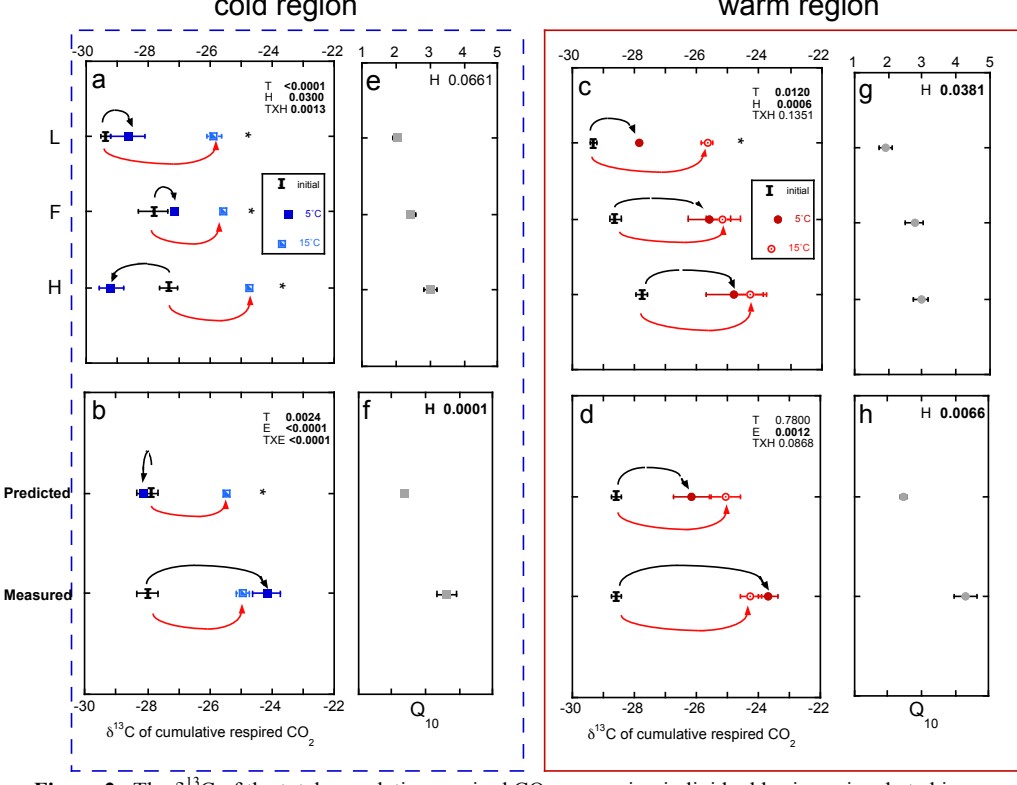

**Figure 2.** The $\delta^{13}C$ of the total cumulative respired $CO_2$ comparing individual horizons incubated in
isolation (isolated experiment; L, F and H; a,c) and whole-profile values (b,d) predicted from those
isolated horizons (predicted) to those measured directly as a whole-profile (measured) with the
corresponding temperature sensitivity ($Q_{10}$; e,f,g,h) of cumulative respiration for the soils from both the
cold and warm regions. Values are given as the mean of three sites ± standard error with the initial bulk
soil $\delta^{13}C$ included for reference. The effect of temperature (T), horizon (H) and their interaction term (T x
H) are given for all three treatments with significance ($\alpha = 0.05$) denoted in bold. Within horizon effect
of T is denoted with an asterisk (*; a,c). The significant effect of T within experiment is denote with an
asterisk (*; b,d).



the initial values and the 5˚C incubations (Fig. 2a). These results indicate that the whole profile structure enhanced the respiration of more $^{13}$C-enriched substrates, to an extent similar to that observed in the warmer incubations. Furthermore, the respiration of more $^{13}$C-enriched substrates was associated with higher $Q_{10}$ of soil respiration regardless of climate region.

Though initial soil organic matter $\delta^{13}$C and $\delta^{15}$N differed by climate region, the changes in soil $\delta^{13}$C and $\delta^{15}$N over the course of the incubation were relatively small (typically less than 1‰) across all horizons, and with both temperatures and experimental types (Table S2). No temperature, horizon or experiment effect was observed in the changes in bulk soil of $\delta^{13}$C and $\delta^{15}$N in these soils.

**3.4 Degradation of soil nitrogen occurred to a greater extent in the cold relative to the warm region soils with no evidence of an effect of whole profile structure.**

The magnitude of change in C:N (ΔC:N) over the course of the experiment decreased with soil depth regardless of region (Fig. 3). Increased temperature resulted in a greater decrease in soil C:N which exhibited decreases of 3-14 and 0-9 in the cold and warm regions, respectively, over the course of the entire incubation (Fig. 3). In the cold region, where the soil C bioreactivity is generally greater (Laganiere et al. 2015), this trend was similar across both the predicted and measured whole experiments, suggesting that soil profile structure was not an important factor for this variable. However, in the warm region soils, the decrease in C:N was evident in the individual L and H horizons as well as the whole profile experiment but not when calculated as a whole profile from the individual horizons. Although a temperature effect was noted within the isolated and both whole treatments of the cold and warm region soils, no effect of soil profile treatment was detected for the ΔC:N within either region. Changes in %C as THAA were only noted in the F sub-horizon of the warm region soils incubated at 15˚C (Fig. S1). No effects of temperature, horizon or experiment were observed for changes in %C as THAA. Changes in %N as THAA were variable, but exhibited temperature effects consistent with the ΔC:N in the cold climate soils in the whole profile soils whether incubated as a whole profile or predicted from the isolated horizons (Fig. 4). Changes in mol% glycine were also consistent with these observations of soil organic N processing in the cold region soil (Fig. S2). The mol% glycine, an indicator of greater microbial reworking of soil organic N including in these soils (Dauwe and Middelburg, 1998; Hedges et al., 1994; Philben et al., 2016), increased with temperature in the L and H sub-horizons as well as the whole organic profiles from the cold region. In the warm region soils, temperature effects were only noted in the isolated experiment where decreases in mol% glycine were observed in the 5˚C relative to 15˚C incubation temperatures (p = 0.0028). This temperature effect was observed within the H sub-horizon (p = 0.0058), indicating a relative increase in final mol% glycine following incubation at the higher relative to lower incubation temperature.



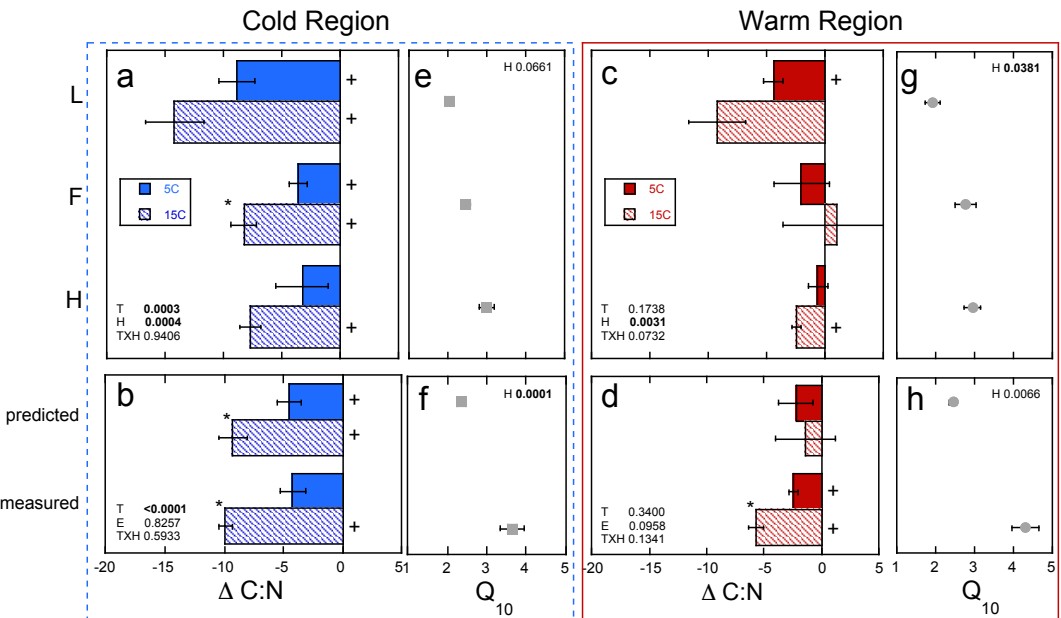

**Figure 3.** The change in soil C:N$_{molar}$ (Δ C:N) given as the final minus initial values comparing the experiment where individual horizons were incubated in isolation from each other (isolated experiment) to both the calculated whole profile values based upon those isolated horizon results (predicted whole experiment) and the actual measured incubation results for whole organic profiles (whole experiment) of cumulative respiration for the soils from both the cold (a,b) and warm regions (c,d) with the corresponding temperature sensitivity (Q$_{10}$; e-g) for reference. Values provided are the mean of the three sites ± standard error with a significant change from 0 denoted by symbol "+" (a-d). For the effect of temperature (T), horizon (H) and their interaction term (T x H) significance (α ≤ 0.05) is denoted in bold. Significance of the effect of temperature within horizon or within experiment is denoted by an asterisk.



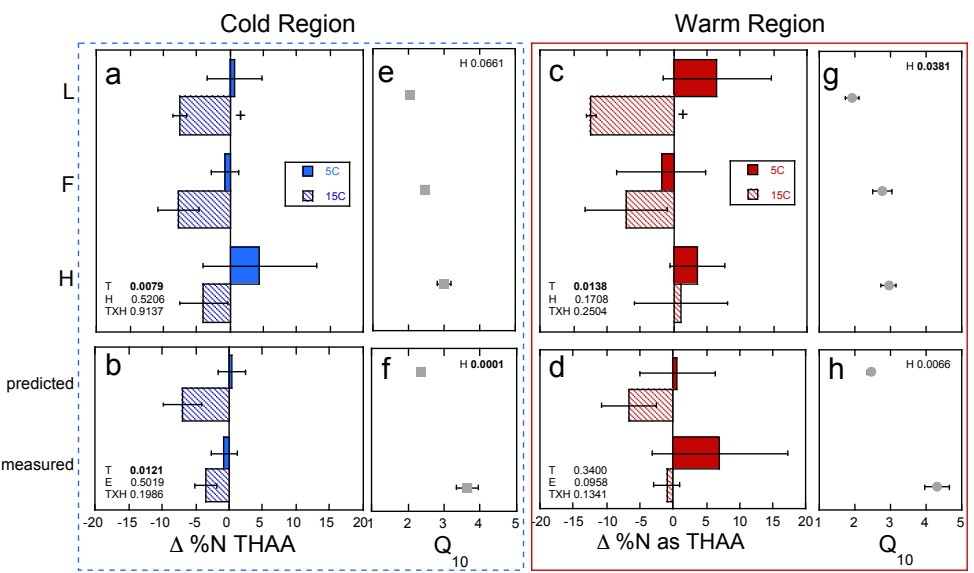

**Figure 4.** The change in %N as total hydrolyzable amino acid (Δ %N as THAA) given as the final minus initial values comparing the experiment where individual horizons were incubated in isolation from each other (isolated experiment) to both the calculated whole profile values based upon those isolated horizon results (predicted whole experiment) and the actual measured incubation results for whole organic profiles (whole experiment) of cumulative respiration for the soils from both the cold (a,b) and warm regions (c,d) with the corresponding temperature sensitivity ($Q_{10}$; e-g) for reference. Values provided are the mean of the three sites ± standard error with a significant change from 0 denoted by symbol "+" (a-d). For the effect of temperature (T), horizon (H) and their interaction term (T x H) significance ($\alpha \leq 0.05$) is denoted in bold. Significance of the effect of temperature within horizon or within experiment is denoted by an asterisk.

### 3.5 Change in soil carbon chemistry during incubation was affected by both temperature and whole soil profile structure

Changes in the alkyl-C to O-alkyl-C ratio (ΔA:O-A) were observed within the L horizons from both climate regions. However, temperature effects on the ΔA:O-A (p = 0.0087) were noted only in the warm region soils and horizon effects (p = 0.021) only in the cold climate soils (Fig S3). Unexpectedly, %alkyl-C (Fig 5) decreased while %O-alkyl-C (Fig S4) increased in the warm region soils over the incubation at 5 ˚C resulting in the decreased ΔA:O-A observed. Decomposition of vascular plant tissues typically results in increases in A:O-A as a result of losses in carbohydrates (rich in o-alkyl-C) and relative retention of plant aliphatics (rich in alkyl-C) (Preston et al., 2009; 2000). This unexpected finding is perhaps a result of the significant surface inputs of moss tissues rich in structural carbohydrates that can be resistant to decomposition (Turetsky et al., 2008; Hájek et al., 2011; Philben et al., 2018). The cold region soils exhibited a horizon effect on the change in %O-alkyl-C (p=0.026)





and ΔA:O-A (p = 0.021) such that %O-alkyl-C generally increased and A:O-A decreased in the L sub-horizon while %O-alkyl-C decreased and A:O-A increased in the H sub-horizon over the incubation (Fig S4). In the warm region soils, increased incubation temperature appeared to have eliminated these unexpected changes in %alkyl-C, %O-alkyl-C and the A:O-A. No change in %alkyl-C, %O-alkyl or A:O-A was noted over the course of the 15°C incubation. However, the unexpected trend of decreasing A:O-A and %alkyl-C was not observed in the warm climate soils incubated as whole soil profiles. An effect of both temperature (p=0.0017) and experiment type (i.e., whole vs isolated) (p=0.0096) was noted in the change in %alkyl-C results. This is observed as an increase in %alkyl-C within the whole profile experiment only when incubated at 15°C as opposed to the reduction in %alkyl-C at 5°C and no change at 15°C in the predicted whole profile (Fig. 5). No experimental effects were noted for either the A:O-A (p = 0.067) or %O-alkyl-C (p = 0.143) changes in the whole soil profiles from the warm region. The %di-O-alkyl-C exhibited a decrease only following the 15°C incubation of the L horizons from both climate regions (Fig S5). However, the isolated horizons from the cold region exhibited both a temperature (p = 0.0004) and horizon (p = 0.044) effect, indicating decreases in %di-O-alkyl-C with warming and primarily within the L sub-horizon. Further, a temperature effect (p = 0.0003), observed as a decrease in %di-O-alkyl-C, was also observed in the whole soil profiles from the cold region regardless of experiment type (Fig S6). The relative changes in the other carbon types as detected via [13]C-NMR were generally below detection. However, we did note increases in %aromatic-C over the course of the 5°C incubation of the warm region soils, consistent with the proportional decreases as observed in %alkyl-C (data not shown).

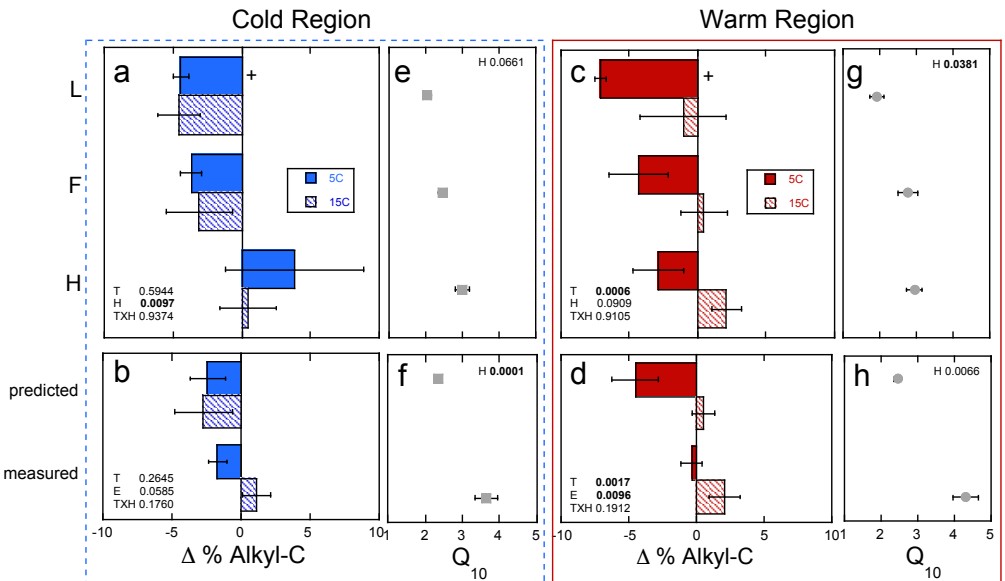

**Figure 5.** The change in the %alkyl-C (Δ %Alkyl-C) given as the final minus initial values comparing the experiment where individual horizons were incubated in isolation from each other (isolated experiment) to both the calculated whole organic profiles values based upon those isolated horizon results (predicted whole experiment) and the actual measured incubation results for whole organic profiles (whole experiment) with the corresponding temperature sensitivity ($Q_{10}$) of cumulative respiration for the soils from both the cold (a-d) and warm regions (e-h). Values provided are the mean of the three sites ± standard error with a significant change from 0 denoted by the symbol "+" (a, e). For the effect of temperature (T), horizon (H) and their interaction term (T x H) significance ($\alpha \leq$ 0.05) is denoted in bold. Significance within horizon or within experiment is denoted by an asterisk, note that no such effects were detected here.





## 4 Discussion

Prior work at these same sites has demonstrated how the temperature response of respiratory C loss from mesic boreal forest organic soils can be enhanced from warmer relative to colder regions, but only when sub-horizons were incubated as a whole soil horizon (Podrebarac et al., 2016). Isolated horizons of the same soils, under the same conditions, exhibited lower respiratory responses to increased temperature, and exhibited no regional climate differences despite clear differences in soil bioreactivity between the two climate regions (Laganière et al., 2015; Podrebarac et al., 2016). These previous studies suggest that some mechanism facilitated by the connectivity among soil horizons can enhance the temperature sensitivity of soil respiration. Here, we explore the potential role of soil priming as a mechanism supporting the enhanced temperature responses of respiration in these whole soil profiles. We addressed two hypotheses probing how soil profile connectivity affects the temperature response of soil respiration by following soil C and N use across soils that differ across depths and regions in SOC and SON composition. First, we found evidence for labile C but not labile N priming of microbial activities, supported by soil horizon connectivity that indirectly stimulates respiratory responses to increased temperature. Second, the data indicate that the labile C priming mechanism is most emergent in the warmer region soils, consistent with lower SOC bioreactivity in the warmer region relative to soil from the colder region.

### 4.1 Enhancement of microbial labile carbon use supports an indirect mechanism for increased temperature sensitivity of soil respiration in whole organic soil profiles

The availability of relatively faster-turnover inputs in incubations that included L sub-horizons with the slower turnover pools within the F and H horizons promoted microbial use of more bioreactive substrates, providing indirect promotion of the enhanced temperature response of soil respiratory C losses in whole soil profiles. Enhanced temperature sensitivity of microbial activity supported indirectly through increased availability of labile substrates, or a priming effect, has been observed with root exudates and other fresh plant inputs (Curiel Yuste et al., 2004) (Zhu and Cheng, 2011) but, to the best of our knowledge, has not yet been linked to soil profile connectivity. These whole soil profiles exhibited increases in the $\delta^{13}C$ of respired $CO_2$ and relative enhancement in the decomposition of carbohydrates compared to soil sub-horizons incubated in isolation. These increases were associated with the enhanced temperature response of respiration in the whole organic profiles not predicted by the response of the same individual horizons incubated in isolation. Most strongly observed in the warm region forest soils, where composition metrics indicated initial lower bioreactivity, the enhanced temperature sensitivity of respiration with the whole soil profile structure appears to be largely supported by labile C priming as we had hypothesized.

Soil connectivity enhanced labile substrate use and the temperature response of soil respiration but lower C loss rates overall. This could be explained by a priming effect within these soil profiles enhancing the use of more complex higher $E_a$ substrates consistent with respiratory temperature responses closer to intrinsic values relative to those of isolated horizons lacking this priming effect (Davidson and Janssens, 2006). Root exudates have been found to increase the availability or use of complex, high $E_a$ substrates via a priming effect (Bingeman et al., 1953; Cheng



et al., 2014), by accelerating SOM decomposition via co-metabolism or the increased production of polymer degrading enzymes breaking down macromolecules and generating more soluble molecules (Schimel and Weintraub, 2003; Wallenstein and Weintraub, 2008; Zhu and Cheng, 2011). Similarly, the whole-soil profiles likely promoted availability of a diversity of substrates and activity of more diverse microbes than in isolated sub-horizons, supporting co-metabolism or increased polymer degrading enzyme activity (Basler et al., 2015) as noted with litter additions (Malik et al., 2016). The lack of change in respiration rates throughout the whole profile incubation, in contrast with horizons incubated in isolation for which respiration rates decreased over time, suggests a maintained substrate availability within the whole profiles (Laganière et al., 2015; Podrebarac et al., 2016). Microbial use of

lower $E_a$ compounds to support microbial responses to increasing temperature, despite lower overall rates of C release in the whole soil profiles relative to that predicted from the isolated sub-horizons, is perhaps a consequence of enhanced substrate assimilation and use efficiency supported by soil connectivity.

The observed increases in $\delta^{13}C$ of respired $CO_2$ and use of carbohydrates here suggests enhanced catabolism of more rapid turnover substrates, likely fueling the use and incorporation of lower $E_a$ compounds in support of increased temperature sensitivity of soil respiration in the whole profile soils. Soil substrates are not uniformly available to microbes, and both temperature and changes in the suite of available substrates can alter the catabolism or incorporation of soil substrates by microbes (Bölscher et al., 2017; Fontaine et al., 2007; Frey et al., 2013; Streit et al., 2014; Zogg et al., 1997). During incubations, catabolism of more recent, fast turnover soil inputs relative to the bulk soil can occur along with an increased proportion of older, slower-turnover soil substrates

incorporated into microbial biomass (Blagodatskaya et al., 2011). The increases in the $\delta^{13}C$ of respired $CO_2$ observed with the whole profile structure, independent of region or incubation temperature, were congruent with increases in $\delta^{13}C$ of respired $CO_2$ observed with incubation temperature in the isolated horizons. These increases exceed those typically attributed to isotopic discrimination associated with respiration (<1‰; (Breecker et al., 2015; Czimczik and Trumbore, 2007) and likely resulted from enhanced use of more $^{13}C$-enriched substrates, increased reuse of bioreactive C pools incorporated into microbial biomass, and/or shifts in microbial composition of the active community (Blagodatskaya et al., 2011).

Enhanced carbohydrate use suggests that catabolism of more bioreactive, $^{13}C$-enriched substrates supported the use of lower $E_a$ substrates, explaining the elevated $Q_{10}$ of soil respiration within the whole profile soils. The relative retention of alkyl-C, likely a consequence of enhanced use of O-alkyl-C, was observed in the warm region

soils, consistent with the increased $\delta^{13}C$ of respired $CO_2$. When the soil horizons were incubated at 5°C as isolated horizons we observed the *opposite* of the typical relative loss of carbohydrates (reduced O-alkyl-C) and retention of plant waxes (retained alkyl-C) associated with the decomposition of vascular plant tissues (Preston et al., 2000; 2009). The unexpected trend of decreasing alkyl-C and A:O-A, observed in the 5°C incubation of the cold and warm region L horizons, was notably absent in the 15°C incubations of warm region soils. The experimental effect on the change in alkyl-C indicated that the whole horizon structure enhanced decomposition more typical of increased carbohydrate use which was further enhanced with increased temperature.

Regional differences in soil composition caused by shifts in input sources associated with longer term climate change is consistent with the more emergent effects of labile C use on respiratory responses to temperature





noted in the warmer region soils. The lack of change in %alkyl-C or %O-alkyl-C within the cold region soils likely

resulted from the larger relative concentration of carbohydrates in the cold region soils, a consequence of the greater moss contributions (Kohl et al., 2018)). The cold region profiles lack the increase in the A:O-A with depth observed in the warm region profiles and typically observed in vascular plant dominated soils including boreal forests (Kane et al., 2010). Therefore, the increases in $\delta^{13}C$-$CO_2$ associated with temperature and the proximity of sub-horizons within the whole profile may still have been due to enhanced mineralization of carbohydrates in the cold region soils not clearly detected in bulk changes in the chemistry of the SOM. This seems likely considering the increases in $\delta^{13}C$-$CO_2$ associated with temperature in the cold region soils occurred across all horizons where decreasing fungal relative to bacterial ratios occurs with depth which can also control the $\delta^{13}C$ of substrate use (Kohl et al., 2015).

**4.2 The microbial consortium supported by soil connectivity may enable an enhanced catabolism of**

**carbohydrate-rich substrates supporting priming effects within the whole soil profile**

By initiating key steps in decomposition of more complex soil organic matter (Paterson et al., 2008), fungi can enhance the decomposition of higher $E_a$ substrates found within the F and H layers, and likely to a greater extent when incubated in contact with the fungal rich L layer. Enhanced fungal enzyme activity is not exclusive of enhanced bacterial respiration and use of carbohydrates in these soil profiles; rather, the observations made in this study suggest co-metabolism resulted in labile substrate use and accelerated complex SOM decomposition supported by the whole profile structure (Kuzyakov 2002). The reduction in respiration and the enhanced respiratory response to temperature could be a result of enhanced fungal activity supported by low bacterial to fungal ratios of the L soils in contact with lower F and H horizons as fungi can exhibit greater substrate use efficiencies (Bölscher et al., 2016; Kallenbach et al., 2016). Enzyme activity and microbial composition is known to vary significantly from L to H

horizons in organic soils (Baldrian et al., 2008; Šnajdr et al., 2008) with fungal biomass attributed to polysaccharide hydrolase activity in surface L horizons in contrast to ligninolytic enzyme activity in deeper H horizons (Baldrian et al., 2011). Therefore, it is possible that soil fungi and their hydrolytic activities may support cross horizon enhancement of substrate use including higher $E_a$ substrates in these organic horizons.

The whole profile structure results in the contact between communities with low bacterial to fungal ratios within carbohydrate rich L horizons with communities exhibiting elevated bacterial to fungal ratios within less carbohydrate rich F and H horizons. This likely contributed to a consortium that enabled enhanced decomposition of carbohydrate-C, increased $\delta^{13}C$ of respired $CO_2$, and the strongest enhancement of carbohydrate decomposition observed in warm climate soils where $Q_{10}$ of soil R increased most strongly with whole profile structure. Enhanced bacterial relative fungal respiration, for example, could explain the increases in $\delta^{13}C$ of respired $CO_2$ observed in the

whole profiles (Dijkstra et al., 2006; Glaser and Amelung, 2002), independent of region or incubation temperature, and congruent with increases observed with incubation temperature in the isolated horizons. This, together with the $\delta^{13}C$ of respired $CO_2$ by horizon and climate region, suggests enhancement in bacterial relative to fungal respiration and bacterial catabolism of carbohydrates occurred in the whole soil profiles.





Temperature enhanced increases in the $\delta^{13}$C-$CO_2$ in all horizons in the cold region forests. However, only the L sub-horizon in the warm region forest soil supported the enhancement of bacterial relative to fungal activity and consequently differences in temperature impacts on substrate use between soils from the two climate regions. Previous investigations indicate a greater proportion of fungal biomass in the cold relative to the warm climate forest soils driven largely by differences in the L sub-horizon (Kohl et al., 2020) consistent with climate trends of increasing bacterial to fungal composition with warming (Pietikainen et al., 2005) as well as enhanced fungal

biomass in winter (Buckeridge et al., 2013; Schadt et al., 2003). Substrate use by soil fungi and bacteria can differ (Koranda et al., 2014; Rinnan and Baath, 2009), and a previous study of the soils from across this same climate transect indicates increased bacterial use of more labile C and fungal use of more slow-turn over pools with warming (Ziegler et al., 2013) consistent with enhanced bacterial relative to fungal activity with warming noted in other studies (Bell et al., 2009; Lipson et al., 2002). The overall increase in $\delta^{13}$C of respired $CO_2$ with depth from the L through H soil horizons incubated in isolation is consistent with increases in $\delta^{13}$C of substrates as well as the increased proportion of bacteria relative to fungi with depth in the organic soils from these forests (Kohl et al., 2015). Therefore, the greater temperature induced increase in $\delta^{13}$C of respired $CO_2$ may indicate an enhancement of bacterial respiration within the cold climate forest horizons where fungi were initially more dominant. This may be why we observed a consistent temperature response across all soil horizons in the cold forest soils, but only in the

upper horizons (where fungi predominate) from the warm region forests.

### 4.3 Enhanced temperature sensitivity of respiration is not associated with enhanced N use in whole soil profiles

We additionally hypothesized that enhanced soil N availability supported through soil connectivity, via the whole soil profile structure, has the potential to enhance the temperature response of soil respiration given horizon differences in soil N content and microbial community composition. Soil N content differed by climate region in these forests, a feature largely attributed to elevated N cycling in the warm region forests (Philben et al., 2016), providing us with an opportunity to assess the role of soil N exchange and use across a climate relevant range of availabilities. Soil N availability can impact the soil microbial community, its substrate use and growth efficiency (Blagodatskaya et al., 2014; Mooshammer et al., 2014), and priming effects supported by fungi (Dijkstra et al.,

2013). Substrate use could therefore be influenced by exchange across horizons as both fungal abundance and soil organic N concentration and composition vary with depth in most soil profiles and particularly in the boreal forest organic horizons explored here (Kohl et al., 2015; Philben et al., 2016). In fact greater N-rich organic substrate availability with increased warming enhanced fungal and phenoloxidase activity (Li et al., 2013) suggesting N could play a role in controlling substrate use and its response to temperature in these soils.

Despite the observed temperature effects on soil N losses consistent with enhanced $N_2O$ production with warming observed in these organic horizons (Buckeridge et al. 2020), and clear differences in availability of soil N represented by the two climate region soils, we observed no difference in soil N use between soils incubated as a whole profile versus as isolated horizons. In particular the lower N availability of the cold region soils, indicated by lower initial %N and higher C:N, suggests that if we were to see an enhancement in soil organic N use it would





likely have been observed in the cold region soils as noted. However, the enhanced use of soil organic N in the
colder region with increased incubation temperature was not impacted by whether soil horizons were incubated in
isolation or connected as a whole profile. The observed changes in %N, C:N, and indicators of amino acid
degradation (%N as THAA, mol% glycine) were similar in both soil profile treatments of the cold region soils.
Congruent with the greater bioreactivity and soil C losses observed in the cold relative to warm region soils
(Laganière et al., 2015), decreases in soil C:N were primarily observed in the cold region soils where soil C:N was
initially higher and temperature effects were noted regardless of soil incubation type (isolated horizons or as a whole
profile). This contrasted with the warm region soils where the change in C:N only differed with incubation
temperature in the whole profile treatment.

We observed some increased degradation of amino acids in the isolated horizons during the incubation. For

example, decreases in %N as THAA were detected in the isolated L horizons from both the warm and cold region
indicating we were able to detect soil N use, and found that the greater availability of N in the surface L horizons
supports enhanced N use with increased temperature but without necessarily impacting overall N use in whole
profiles. These results suggest that the enhanced temperature sensitivity of soil respiration observed in the whole
profile soils relative to the sum of their isolated horizons is not due to changes in N substrate use but rather C
substrate use facilitated by the whole profile structure.

### 4.4 Conclusions

By demonstrating how whole soil profile structure can impact soil C substrate use in ways that affect
temperature responses of soil $CO_2$ effluxes, this study provides a demonstration of the importance of soil
connectivity as a regulator of soil-climate feedbacks. The degree to which this mechanism exerts itself in other soils

remains unknown, but these results highlight the importance of understanding mechanisms that operate in intact soil
profiles – only rarely studied – in regulating respiratory responses to changing temperature. We demonstrate how
whole profile attributes can regulate soil C cycling and thereby respiratory responses beyond bulk chemical
composition and Arrhenius theory. Given that recent photosynthates, e.g. root exudates, can contribute similarly to
soil respiration across surface to deep horizons (Pumpanen et al., 2009) the role of a priming mechanism suggested
by our study is worthy of investigating in deeper mineral soil profiles where enhanced temperature sensitivity of soil
respiration is supported by soil C of recent origin (Hicks Pries et al., 2017). Deeper soils are certainly key to
uncovering the full soil response to climate change, however, understanding controls on those responses requires an
understanding of cross-horizon exchange processes. Root inputs and hydrologic regimes transferring dissolved
organic matter and nutrients as well as regulating redox conditions represent relevant factors likely controlling

interactive effects of cross-horizon exchange on soil C use, and therefore need to be better integrated in support of
our understanding of soil C use and its response to temperature.



*Appendices*. Supplemental material related to this article available online.

*Data availability*. All data are included in the paper tables and Supplement.

**Author contribution**

640 Authors JL, SB, and SZ contributed to the general conceptions of the study. SB, KE, JL, and SZ designed the sampling. KE, JL, and SZ contributed to field sample collections while incubation experiment set up and sampling was conducted by FP and JL. FP, JL and MN contributed to sample analyses including soil $CO_2$ fluxes, sample extractions and preparations for isotope and elemental analyses, NMR and amino acids. FP conducted data and statistical analyses. FP and SZ jointly wrote the manuscript which received edits from all co-authors.

**Competing interests**

The authors declare that they have no conflict of interest.

650 **Acknowledgements**

We thank Darrell Harris, Andrea Skinner, and Thalia Soucy Giguere for fieldwork assistance also, Rachelle Dove, Geert Van Biesen, Jamie Warren and Catie Young for laboratory assistance.  NMR analyses were conducted by Dr. Céline Schneider in the Centre for Chemical Analyses and Training at Memorial University. Funding was generously provided by NSERC-CRSNG (SPG#479224; DG#2018-05383); NSERC CREATE Programme; Canadian Forest Service, Natural Resources Canada; Forestry and Agri-Foods Agency, Government of NL; Canada Research Chairs Programme.

**Conflicts of interest.**

The authors declare that they have no conflicts of interest or competing interests.

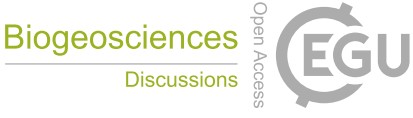

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
