# Peer review of "Soil profile connectivity can impact microbial substrate use, affecting how soil CO2 effluxes are controlled by temperature"

_Biogeosciences, 2020_

## Referee Comment (RC1) · Anonymous Referee #1 · 16 Dec 2020

General comments The authors describe two soil incubation experiments, (1) of individual sub-horizons of boreal forest organic layers and (2) of entire organic layer profiles with their natural sequence from less to more degraded organic matter. Based on a comparison of these two treatments in two forest sites of different climate and at two incubation temperatures, the authors draw conclusions on the impact of exchange processes between shallower and deeper parts of the organic layer on carbon and nitrogen cycling, in particular soil respiration. I find the study interesting and relevant and I agree with the authors that soil connectivity might have a strong (and understudied) effect on biogeochemical processes especially in the deeper soil. However, I have to admit that I cannot follow the argument in the discussion; this includes both difficulties

understanding the meaning of the partly convoluted text and relating the interpretation to measured parameters. I recommend to carefully revise the discussion, refer to figures and tables throughout and consider an overview figure or scheme.

Specific comments Figures 3, 4, 5: The Q10 values seem oddly similar in the three figures and I can't align them with the temperature comparisons. Is it possible that you forgot to change this part of the figure moving from one to the other? Also, what do you mean with "cumulative respiration" in the legends of these figures (that are not about respiration)? Sub-chapter 4.1: I have difficulties following this central argument. If you are referring to actual respiration data – are they shown anywhere in the manuscript or are they described in a previous publication? If they are in previous publications, would it be possible to show these data here (citing the original publications)? Or do you mean the mass loss data? Is the change in carbohydrates shown in Figure S4; and if Fig. S4 is central to the interpretation, why is it not in the main manuscript? It seems to me that you are describing an increase in both labile and complex substrate degradation by soil connectivity – but are they not measured relative to each other? I understand that the dataset has a complex structure so I tried several times, but I still do not understand this argument. I would also appreciate if you could refer to Figures and Tables in the Discussion to make it easier to follow. Line 558: Soil R? Line 587: What do you mean with "a climate relevant range of availabilities"?

Technical corrections Line 162: "Provide" instead of "provides". Figure 2: The two types of blue squares are difficult to distinguish.

---

## Referee Comment (RC2) · Jorge Curiel Yuste (Referee) · 29 Jan 2021

The study rises very interesting hypotheses (priming is not only triggered by exudates but also by inter-layer exchange of C sources, or priming involves mainly C and not N exchange). I think the experimental design is very smart and well designed to test these hypotheses and I think this might be a great contribution to understand a poorly understood process with, perhaps, large implications in terrestrial C cycling

That said, I must say that following the story-line of this manuscript has been challenging for different reasons. First of all, the experimental design is complex and needs some re-writing, trying to make life easier to readers that perhaps are not familiarized

with former studies produced by this laboratory. Below some suggestions. Second, wording is sometimes a bit chaotic. The feeling is that the first author of the MS has the hypotheses, results and interpretation in the mind, but still needs to improve the way results and interpretations are reported so readers also understand well the mechanisms involved in this inter-layer priming. Third, I think the study will benefit from some reorganization of results and conclusions (see below some suggestions). Fourth, and even if I think that the experimental design is correct and well justified by the hypotheses, I think that authors should also justified how these results can be extrapolated to field processes, since, e.g. the manipulations of the layers in the experiment (e.g. L layer has been homogenized and pieces has been cut) has for sure huge impact over the functioning of the system. Also, the choice in the length of the experiment (more than a year) should be justified in order to understand how this helped testing the hypotheses. Finally, I think that the conclusion section might be rewriting to really synthesize the results obtained based on the hypothesis launched and explaining potential implications of this identified process on terrestrial C cycling. Specific comments - Introduction: too long, difficult to follow the line of arguments. For instance, the whole paragraph 4 (lines 87 to 106) is key to understand the whole study, but understanding it is extremely challenging. I encourage authors to rewrite it. In general. I think that it can also be shortened substantially by integrating better the ideas instead of fragmenting them into different paragraphs (8 in total, too much!)

- Experimental design is poorly explained o The study is built based on results reported in former manuscripts and even if the author's made the effort to explain what was done in the past, it is still difficult to follow a study that built on former studies. I suggest to use a small scheme of the experimental design, showing also the two different approaches (isolated and whole experiment) and the tubes used. This will greatly help readers to understand how this study have put together results reported in other studies o Poor explanations lead also to potential misinterpretations. For instance, it is not clear whether the total amount of soil used in isolated and whole experiments was similar. This should be well explained because If not, it might be interpreted that higher C

losses of "modeled" versus "predicted" comes from the fact that isolated soil layers might experience more oxidation and C losses because surface/volume ratios differ and diffusivity of O2 and CO2 changes due to layer thickness o Author's assume readers knows very well what the alkyl C and O alkyl C or the THAA fractions of the total carbon or total nitrogen pools means, process-based. Why those fractions and their ratios were used in this study to interpret results should be well explained in the materials and method section. On top of that, it is clear that both fractions of the C pool (alkyl C and O alkyl C) and their ratio are very important to understand results, but only part of the results are presented in the MS while the other half has been included only in supplementary materials, which makes also difficult to follow the argumentation

- Results section. I think the result section might be improved too. For instance, and parallel to Figure 3 (initial values for each horizon) it might help readers to understand the story-line and justification of results to show a Figure where to see the absolute changes in the different fractions/ratios shown in Fig 1. This is a part of the results complementary to the relative changes shown in Figs 3-5 that may help understand how the different fractions has been depleted and where. For the same reason, results from Fig S4 seems to give complementary information to those obtained in Fig 5. But to follow the argumentation you need to switch from the MS and the supplementary material.

- Discussion section is too large. Actually I've identified a whole section (section 4.2) that seems utterly speculative with no data on fungal or bacterial activity available in this experiment., I am sure that the whole section can be reduced to a couple of sentences. Overall, the feeling is that the discussion can be substantially reduced in length

- Conclusions. Conclusion section should be re-written to show better the main findings and the potential consequences for the terrestrial C cycling of identifying this process. The study will also gain from including and extra Figure synthesizing the whole complexity of the study into a Figure that explains the mechanisms identified based on the different results obtained.

---

## Author Comment (AC1) · 26 Feb 2021

Here we provide the separate parts from the original review and in each case give our response following "Response:"

General comments The authors describe two soil incubation experiments, (1) of individual sub-horizons of boreal forest organic layers and (2) of entire organic layer profiles with their natural sequence from less to more degraded organic matter. Based on a comparison of these two treatments in two forest sites of different climate and at two incubation temperatures, the authors draw conclusions on the impact of exchange processes between shallower and deeper parts of the organic layer on carbon and nitrogen cycling, in particular soil respiration. I find the study interesting and relevant and I agree with the authors that soil connectivity might have a strong (and understudied) effect on biogeochemical processes especially in the deeper soil. However, I have to admit that I cannot follow the argument in the discussion; this includes both difficulties understanding the meaning of the partly convoluted text and relating the interpretation to measured parameters. I recommend to carefully revise the discussion, refer to figures and tables throughout and consider an overview figure or scheme.

Response: Thank you for your careful consideration of this manuscript and these insights. We will revise taking care to clarify the main argument within the discussion through more explicit reference to the research findings. We agree that an overview figure – likely one that can be used to both highlight the original hypotheses and then help describe the resulting outcome of the study.

Specific comments Figures 3, 4, 5: The Q10 values seem oddly similar in the three figures and I can't align them with the temperature comparisons. Is it possible that you forgot to change this part of the figure moving from one to the other? Also, what do you mean with "cumulative respiration" in the legends of these figures (that are not about respiration)?

Response: The Q10 values are in fact the same values in each of the four cases (2 experimental treatments X 2 climate regions). They were provided this way to maintain that reference to the differences (or lack thereof) among horizons or treatments when looking at the soil chemical changes expressed in these 3 different figures. The Q10 values given in each figure refers to the temperature sensitivity of the cumulative respiration from across the whole experiment so matching with the time frame over which the change in soil C:N (Fig. 3), %N as THAA (Fig. 4), and Alkyl-C (Fig. 5) were observed in these soil incubations. Obviously, this was not clear so we will be sure to clarify both of these issues explicitly within the figure legend of each.

Sub-chapter 4.1: I have difficulties following this central argument. If you are referring

to actual respiration data – are they shown anywhere in the manuscript or are they described in a previous publication? If they are in previous publications, would it be possible to show these data here (citing the original publications)? Or do you mean the mass loss data? Is the change in carbohydrates shown in Figure S4; and if Fig. S4 is central to the interpretation, why is it not in the main manuscript? It seems to me that you are describing an increase in both labile and complex substrate degradation by soil connectivity – but are they not measured relative to each other? I understand that the dataset has a complex structure so I tried several times, but I still do not understand this argument.

Response: Section 4.1 describes the evidence for enhanced use of labile substrates in the whole soil profiles supporting the hypothesis that the whole profile can enhance the temperature response of microbial respiration via a priming effect. The reviewer's comment indicates that this section is not clear and we will seek to edit thoroughly in order to clarify the main point and how it is supported by the findings in this study. This will include more direct reference to the soil respiration data which is in fact previously published. In that case we will consider the best way to present that information citing the publication. We will also consider bringing the supplemental Figure S4 up into the main body of the paper. Clarification through more direct reference to the findings in this study will be made while trying to simplify the discussion of this main point. I would also appreciate if you could refer to Figures and Tables in the Discussion to make it easier to follow.

Response: As indicated with the specific comment on Section 4.1 we will keep this in mind and provide such references to Figures and Tables where it will help the reader follow the key points.

Line 558: Soil R?

Response: That was meant to refer to soil respiration. We will be sure to spell it out in each case to avoid confusion.

Line 587: What do you mean with "a climate relevant range of availabilities"?

Response: Here we are referring to the significant differences in soil organic nitrogen availability (measured as N turnover; Philben et al. 2016) captured from across the regional climate transect studied. This transect represents a range in mean annual T and mean annual precipitation congruent with predictions in climate change for this region over the next century. Thus we are referring to differences in soil N availability relevant to the different climates experiences in these forests. We will edit this to clarify.

Technical corrections

Line 162: "Provide" instead of "provides".

Response: Will fix.

Figure 2: The two types of blue squares are difficult to distinguish.

Response: Will fix.

---

## Author Comment (AC2) · 26 Feb 2021

Here we provide our responses to each part of the reviewer's comments below and following "Response:"

The study rises very interesting hypotheses (priming is not only triggered by exudates but also by inter-layer exchange of C sources, or priming involves mainly C and not N exchange). I think the experimental design is very smart and well designed to test these hypotheses and I think this might be a great contribution to understand a poorly understood process with, perhaps, large implications in terrestrial C cycling That said, I must say that following the story-line of this manuscript has been challenging for

different reasons. First of all, the experimental design is complex and needs some re-writing, trying to make life easier to readers that perhaps are not familiarized with former studies produced by this laboratory. Below some suggestions. Second, wording is sometimes a bit chaotic. The feeling is that the first author of the MS has the hypotheses, results and interpretation in the mind, but still needs to improve the way results and interpretations are reported so readers also understand well the mechanisms involved in this inter-layer priming. Third, I think the study will benefit from some reorganization of results and conclusions (see below some suggestions). Fourth, and even if I think that the experimental design is correct and well justified by the hypotheses, I think that authors should also justified how these results can be extrapolated to field processes, since, e.g. the manipulations of the layers in the experiment (e.g. L layer has been homogenized and pieces has been cut) has for sure huge impact over the functioning of the system. Also, the choice in the length of the experiment (more than a year) should be justified in order to understand how this helped testing the hypotheses. Finally, I think that the conclusion section might be rewriting to really synthesize the re- sults obtained based on the hypothesis launched and explaining potential implications of this identified process on terrestrial C cycling.

Response: Thank you for your careful evaluation of this manuscript and the constructive critique which will guide our revisions and greatly improve this paper.

Specific comments – Introduction: too long, difficult to follow the line of arguments. For instance, the whole paragraph 4 (lines 87 to 106) is key to understand the whole study, but understanding it is extremely challenging. I encourage authors to rewrite it. In general. I think that it can also be shortened substantially by integrating better the ideas instead of fragmenting them into different paragraphs (8 in total, too much!)

Response: We will work on integrating the main ideas covered in the introduction section and shorten that section.

Experimental design is poorly explained o The study is built based on results reported

in former manuscripts and even if the author′s made the effort to explain what was done in the past, it is still difficult to follow a study that built on former studies. I suggest to use a small scheme of the experimental design, showing also the two different approaches (isolated and whole experiment) and the tubes used. This will greatly help readers to understand how this study have put together results reported in other studies. Poor explanations lead also to potential misinterpretations. For instance, it is not clear whether the total amount of soil used in isolated and whole experiments was similar. This should be well explained because If not, it might be interpreted that higher C losses of "modeled" versus "predicted" comes from the fact that isolated soil layers might experience more oxidation and C losses because surface/volume ratios differ and diffusivity of O2 and CO2 changes due to layer thickness

Response: It is clear from this feedback in addition to the other reviewer's feedback that we relied too heavily upon the previous publication and in doing so have left out key information necessary to accurately communicate the findings of this study. We agree a figure illustrating the experimental design would help in this regard. For example, such a figure and a bit more detail will help convey that the total amount of soil used within the whole and isolated experiments was in fact similar and the surface/volume ratios in each constrained by the use of the same microcosm tubes. We will provide a new figure and revise the methods section to provide more information required to clarify the experimental design.

Author′s assume readers knows very well what the alkyl C and O alkyl C or the THAA fractions of the total carbon or total nitrogen pools means, process-based. Why those fractions and their ratios were used in this study to interpret results should be well explained in the materials and method section.

Response: Agreed, we will add an explanation within the Methods section that ties each of the measurements made to the hypotheses being tested.

On top of that, it is clear that both fractions of the C pool (alkyl C and O alkyl C) and

their ratio are very important to understand results, but only part of the results are presented in the MS while the other half has been included only in supplementary materials, which makes also difficult to follow the argumentation

Response: This is a similar concern raised by the other reviewer. It is abundantly clear that pulling the O-Alkyl-C results figure into the main body of the paper would be helpful in clarifying the results and key findings of this study. We will do that.

Results section. I think the result section might be improved too. For instance, and parallel to Figure 3 (initial values for each horizon) it might help readers to understand the story-line and justification of results to show a Figure where to see the absolute changes in the different fractions/ratios shown in Fig 1. This is a part of the results complementary to the relative changes shown in Figs 3-5 that may help understand how the different fractions has been depleted and where. For the same reason, results from Fig S4 seems to give complementary information to those obtained in Fig 5. But to follow the argumentation you need to switch from the MS and the supplementary material.

Response: This comment and the previous make it clear that it would be easier on the reader were we to simply provide all 6 of the same datasets as expressed in the 6 panels of Fig.1 rather than just the three provided in Figs. 3-5. We will amend the results to reflect that change as well as organizing around the three key N based measures and the three key C based measures. If possible this might be done by constructing two new figures each with three sets of panels providing those absolute changes as recommended. Either way we will incorporate these figures into the main text as suggested here.

Discussion section is too large. Actually I′ve identified a whole section (section 4.2) that seems utterly speculative with no data on fungal or bacterial activity available in this experiment., I am sure that the whole section can be reduced to a couple of sentences. Overall, the feeling is that the discussion can be substantially reduced in

length

Response: We will edit the discussion substantially to excise extraneous discussion points and better focus on the main point that the study supports and the potential implications.

Conclusions. Conclusion section should be re-written to show better the main findings and the potential consequences for the terrestrial C cycling of identifying this process. The study will also gain from including and extra Figure synthesizing the whole complexity of the study into a Figure that explains the mechanisms identified based on the different results obtained.

Response: We will revise the conclusion section to clarify the main finding and implication. We will also generate a conceptual figure that addresses the point you raise here and anticipate that figure may also be constructed to aid us in clarifying the hypotheses posed, how they link to the measures made, and at the same time clarifying the main findings.

---

## Author Response (AR1)

**Response to Reviews**

Referee comments in are in *italics* and are followed by our response and specific edits made in response where appropriate. All line numbers refer to the final revised manuscript version without tracked changes.

Anonymous Referee #1

*General comments The authors describe two soil incubation experiments, (1) of individual sub-horizons of boreal forest organic layers and (2) of entire organic layer profiles with their natural sequence from less to more degraded organic matter. Based on a comparison of these two treatments in two forest sites of different climate and at two incubation temperatures, the authors draw conclusions on the impact of exchange processes between shallower and deeper parts of the organic layer on carbon and ni- trogen cycling, in particular soil respiration. I find the study interesting and relevant and I agree with the authors that soil connectivity might have a strong (and understudied) effect on biogeochemical processes especially in the deeper soil. However, I have to admit that I cannot follow the argument in the discussion; this includes both difficulties understanding the meaning of the partly convoluted text and relating the interpretation to measured parameters. I recommend to carefully revise the discussion, refer to figures and tables throughout and consider an overview figure or scheme.*

**Response:** Thank you for your careful consideration of this manuscript and these important insights which have guided our revisions. We have revised taking care to clarify the main argument within the discussion through more explicit reference to the research findings. We provided an overview figure that provides background and helps to describe the original hypotheses of this study.

**Edits made:** The discussion has been edited to shorten and clarify throughout including reorganization of section 4.1 to clarify the main points regarding the role of labile C priming supporting the enhanced temperature sensitivity of soil respiration. Section 4.2 has been removed and 6 sentences describing how the varied microbial communities of the L, F and H horizons incubated as a whole horizon may help explain the observed patterns in substrate use have been incorporated into the

previous section (see Lines 559-572). These edits have resulted in over a 20% reduction in the length of the discussion and we hope have clarified the main points of discussion.

Specific comments

*(1) Figures 3, 4, 5: The Q10 values seem oddly similar in the three figures and I can't align them with the temperature comparisons. Is it possible that you forgot to change this part of the figure moving from one to the other? Also, what do you mean with "cumulative respiration" in the legends of these figures (that are not about respiration)?*

**Response:** The $Q_{10}$ values are in fact the same values in each of the four cases (2 experimental treatments X 2 climate regions). They were provided this way to maintain that reference to the differences (or lack thereof) among horizons or treatments when looking at the soil chemical changes expressed in these 3 different figures. The $Q_{10}$ values given in each figure refers to the temperature sensitivity of the cumulative respiration from across the whole experiment so matching with the time frame over which the change in soil C:N (Fig. 3), %N as THAA (Fig. 4), and Alkyl-C (Fig. 5) were observed in these soil incubations. Obviously, this was not clear so we edited to clarify both of these issues explicitly within the figure legend of each.

**Edited steps taken:** Figures 3, 4 and 5 were combined alongside supplemental figures S2, S3 and S4 to provide two new figures that depict the soil C and N metrics assessed and reduces the redundancy and clarifies the comparison to the $Q_{10}$ of respiration. This arrangement of the data also helps to create better links between the hypotheses, initial soil properties and these findings. We have also edited the caption for these two figures to more clearly describe what the $Q_{10}$ values refer to including the term cumulative respiration.

*Sub-chapter 4.1: I have difficulties following this central argument. If you are referring to actual respiration data – are they shown anywhere in the manuscript or are they described in a previous publication? If they are in previous publications, would it be possible to show these data here (citing the original publications)? Or do you mean the mass loss data? Is the change in carbohydrates shown in Figure S4; and if Fig. S4 is central to*

*the interpretation, why is it not in the main manuscript? It seems to me that you are describing an increase in both labile and complex substrate degradation by soil connectivity – but are they not measured relative to each other? I understand that the dataset has a complex structure so I tried several times, but I still do not understand this argument.*

*I would also appreciate if you could refer to Figures and Tables in the Discussion to make it easier to follow.*

**Response:** Section 4.1 describes the evidence for enhanced use of labile substrates in the whole soil profiles supporting the hypothesis that the whole profile can enhance the temperature response of microbial respiration via a priming effect.  The reviewer's comment indicates that this section is not clear so we edited thoroughly in order to clarify the main point and how it is supported by the findings in this study.  This includes a more direct reference to the soil C losses measured within this study and inclusion of a supplemental figure (Fig. S3) depicting some of the previously published respiration data. We also incorporated the NMR data depicting alkyl to O-alkyl C within a newly organized figure so that the data on changes in carbohydrate and labile C composition is more clearly depicted in the main body of the paper (see new Fig. 5). Clarification through more direct reference to the findings in this study is now made while trying to simplify the discussion of this main point.

**Edits made:** The discussion section 4.1 has been edited to address the reviewer's points here (see Lines 546-558).  First, we've clarified that soil C losses (Table 1) and respiration rates (Fig. S3) were lower in the whole relative to isolated soil experiments. The supplemental figure (Fig. S3) has been added providing the previously published respiration data to clarify the supporting points made here in relationship to the soil compositional changes presented in this study. Second, editing and reorganization of this section was completed to help the better support the evidence presented regarding the labile C priming effect and its support of the temperature response of soils respiration. This included reference, where appropriate, to figures depicting key results. This changes enabled a much clearer description of the main points of this section.

*Line 558: Soil R?*

**Response:** That was meant to refer to soil respiration. We will be sure to spell it out in each case to avoid confusion.

**Edits made:** replaced R with respiration

*Line 587: What do you mean with "a climate relevant range of availabilities"?*

**Response:** Here we are referring to the significant differences in soil organic nitrogen availability (measured as N turnover; Philben et al. 2016) captured from across the regional climate transect studied. This transect represents a range in mean annual T and mean annual precipitation congruent with predictions in climate change for this region over the next century. Thus we are referring to differences in soil N availability relevant to the different climates experiences in these forests. We have edited this to clarify.

**Edits made:** New text now reads "Soil N content differed by climate region in these forests, a feature largely attributed to elevated N cycling in the warm region forests (Philben et al., 2016), providing us with an opportunity to assess the role of soil N exchange and use across a climate relevant range of soil N availability in this boreal forest region."

*Technical corrections*

*Line 162: "Provide" instead of "provides".*

**Edits made:** Fixed.

*Figure 2: The two types of blue squares are difficult to distinguish.*

**Edits made:** Changed both the color and symbol (now completely open) of the 15C colder site indicator on Fig. 2 (now Fig. 4) to make it more distinguishable.

**Jorge Curiel Yuste (referee)**

*The study rises very interesting hypotheses (priming is not only triggered by exudates but also by inter-layer exchange of C sources, or priming involves mainly C and not N exchange). I think the experimental design is*

*very smart and well designed to test these hypotheses and I think this might be a great contribution to understand a poorly understood process with, perhaps, large implications in terrestrial C cycling*

*That said, I must say that following the story-line of this manuscript has been challenging for different reasons. First of all, the experimental design is complex and needs some re-writing, trying to make life easier to readers that perhaps are not familiarized with former studies produced by this laboratory. Below some suggestions. Second, wording is sometimes a bit chaotic. The feeling is that the first author of the MS has the hypotheses, results and interpretation in the mind, but still needs to improve the way results and interpretations are reported so readers also understand well the mechanisms involved in this inter-layer priming. Third, I think the study will benefit from some reorganization of results and conclusions (see below some suggestions). Fourth, and even if I think that the experimental design is correct and well justified by the hypotheses, I think that authors should also justified how these results can be extrapolated to field processes, since, e.g. the manipulations of the layers in the experiment (e.g. L layer has been homogenized and pieces has been cut) has for sure huge impact over the functioning of the system. Also, the choice in the length of the experiment (more than a year) should be justified in order to understand how this helped testing the hypotheses. Finally, I think that the conclusion section might be rewriting to really synthesize the re- sults obtained based on the hypothesis launched and explaining potential implications of this identified process on terrestrial C cycling.*

**Response:** Thank you for your careful evaluation of this manuscript and the constructive critique that has guided our revisions and greatly improve this paper.

**Edits made:** In response to this overall comments; (1) The section describing the experimental design has been edited and now refers to a diagram illustrating the experimental set up which we feel helps a great deal to clarify what was done in this study; (2) We constructed a conceptual figure depicting the previous findings and the current hypotheses being tested in this study.  This figure is referred to within the introduction, methods (where statistics are described) and in the results and discussion.  We feel that this combined with some overall editing of

the writing has improved the clarity of the results are reported and interpreted.; (3) The results and conclusions have been rewritten taking the suggestions given below (see details below); and (4) revisions within the methods and discussion were made to clarify how these findings relate to what we would expect in situ both describing the soil manipulations and timing of the incubations. Here we describe the use of intact cores where soils were not separated and homogeneized as well as the in situ soil respiration studies conducted in the same sites and which support the findings observed in these more controlled experiments (Lines 178-186). Further we describe the reasoning and caveat associated with only capturing the change in soil composition after the entire incubation period (Lines 211-217).

*Specific comments –*

*Introduction: too long, difficult to follow the line of arguments. For instance, the whole paragraph 4 (lines 87 to 106) is key to understand the whole study, but understanding it is extremely challenging. I encourage authors to rewrite it. In general. I think that it can also be shortened substantially by integrating better the ideas instead of fragmenting them into different paragraphs (8 in total, too much!)*

**Response:** We worked on integrating the main ideas covered in the introduction section and shorten that section.

**Edits made:** The original paragraph 4 is now paragraph 3 and has been edited for clarity and shortened as a result (Lines 66-78). The entire introduction was reorganized and shortened such that it is now only 5 paragraphs and about 20% shorter than the original introduction.

*Experimental design is poorly explained o The study is built based on results reported in former manuscripts and even if the author's made the effort to explain what was done in the past, it is still difficult to follow a study that built on former studies. I suggest to use a small scheme of the experimental design, showing also the two different approaches (isolated and whole experiment) and the tubes used. This will greatly help readers to understand how this study have put together results reported in other studies. Poor explanations lead also to potential misinterpretations. For instance, it is not clear whether the total amount of soil used in isolated*

*and whole experiments was similar. This should be well explained because If not, it might be interpreted that higher C losses of "modeled" versus "predicted" comes from the fact that isolated soil layers might experience more oxidation and C losses because surface/volume ratios differ and diffusivity of O2 and CO2 changes due to layer thickness*

**Response:** It is clear from this feedback in addition to the other reviewer's feedback that we relied too heavily upon the previous publication and in doing so have left out key information necessary to accurately communicate the findings of this study. We have included a figure illustrating the experimental design to help convey that the total amount of soil used within the whole and isolated experiments was in fact similar and the surface/volume ratios in each constrained by the use of the same microcosm tubes. We also revised the methods section to provide more information required to clarify the experimental design.

**Edits made:** A new figure (Fig. 2) illustrating the experimental design has been added to the methods section. We feel that this goes a long way in clarifying the experimental design. For example, it illustrates the common soil mass used in both the isolated and whole experiments (now also indicated in the results on new version lines 175-178).

*Author's assume readers knows very well what the alkyl C and O alkyl C or the THAA fractions of the total carbon or total nitrogen pools means, process-based. Why those fractions and their ratios were used in this study to interpret results should be well explained in the materials and method section.*

**Response:** Agreed, we added an explanation within the Methods section that ties each of the measurements made to the hypotheses being tested.

**Edits made:** An explanation for the use of the specific C functional groups and amino acid measures is provided in the methods section. See lines 247-252 and 295-302 in the revision.

*On top of that, it is clear that both fractions of the C pool (alkyl C and O alkyl C) and their ratio are very important to understand results, but only part of the results are presented in the MS while the other half has been included only in supplementary materials, which makes also difficult to*

*follow the argumentation*

**Response:** This is a similar concern raised by the other reviewer. It is abundantly clear that pulling the O-Alkyl-C results figure into the main body of the paper would be helpful in clarifying the results and key findings of this study. We did that through some new figures.

**Edits made:** Original Figure 3 (now Figure 5) has been edited such that in now contains all of the main C compositional results, including absolute changes in C:N, %alkyl-C, %di-O-alkyl C, and the ratio of alkyl C to O-alkyl C (A:O-A) results. This resulted in pulling the data presented in Supplemental figures S3 and S5 into this new Fig. 5.

*Results section. I think the result section might be improved too. For instance, and parallel to Figure 3 (initial values for each horizon) it might help readers to understand the story-line and justification of results to show a Figure where to see the absolute changes in the different fractions/ratios shown in Fig 1. This is a part of the results complementary to the relative changes shown in Figs 3-5 that may help understand how the different fractions has been depleted and where. For the same reason, results from Fig S4 seems to give complementary information to those obtained in Fig 5. But to follow the argumentation you need to switch from the MS and the supplementary material.*

**Response:** This comment and the previous make it clear that it would be easier on the reader were we to simply provide all 6 of the same datasets as expressed in the 6 panels of Fig.1 rather than just the three separate figures original provided in Figs. 3-5. We amended the results to reflect that change as well as organized it around the three key N based measures and the three key C based measures. This was done by constructing two new figures each with a full set of panels providing those absolute changes as recommended.

**Edits made:** Figure 5 is now a complete set of panels that provide all of the same C measures depicted in the initial soil parameters figure by depth. The new figure 6 provides a complete set of panels that provides all of the same N measures, absolute change in %N as THAA and mol% glycine depicted in the initial soil parameter figure by depth. This helps to organized the results in a way that better reflects the changes around

the major C based measures and N based measures.  The methods were edited to reflect these major figure changes.

*Discussion section is too large. Actually I've identified a whole section (section 4.2) that seems utterly speculative with no data on fungal or bacterial activity available in this experiment., I am sure that the whole section can be reduced to a couple of sentences. Overall, the feeling is that the discussion can be substantially reduced in length*

**Response:** We edited the discussion substantially to excise extraneous discussion points and better focus on the main point that the study supports and the potential implications.

**Edits made:** Section 4.2 has been removed and 6 sentences describing how role that varied microbial communities of the L, F and H horizons incubated as a whole horizon may help explain the observed patterns in substrate use have been incorporated into the previous section (see Lines 559-572). This meant retaining only the points directly relevant to supporting the substrate use observations made in this study. Further, the discussion has been edited to shorten and clarify throughout including reorganization of section 4.1 to help clarify that main points regarding the role of labile C priming supporting the enhanced temperature sensitivity of soil respiration.  This has resulted in over a 20% reduction in the length of the discussion.

*Conclusions. Conclusion section should be re-written to show better the main findings and the potential consequences for the terrestrial C cycling of identifying this process. The study will also gain from including and extra Figure synthesizing the whole com- plexity of the study into a Figure that explains the mechanisms identified based on the different results obtained.*

**Response:** We revised the conclusion section to clarify the main finding and implication and generated a conceptual figure that addresses the point you raise here. This same figure also helps to clarify the hypotheses posed, how they link to the measures made.

**Edits made:** The conclusion section has been edited for clarification including a more specific subtitle (see lines 608-623) and reference to the

conceptual figure used to introduce the hypotheses, expected and observed results from this study. We hope that this paragraph more clearly states the main findings and their potential consequences for terrestrial C cycling.

---

## Referee Report (RR1)

[revised manuscript text omitted]

* * *
Margin comments:

Susan Ziegler 2021-3-13 3:15 PM

Susan Ziegler 2021-3-13 9:16 AM
**Moved (insertion) [2]**

Susan Ziegler 2021-3-13 4:02 PM

Susan Ziegler 2021-4-7 11:55 AM

Susan Ziegler 2021-3-13 9:17 AM

Susan Ziegler 2021-3-13 9:16 AM
**Moved up [2]:** Legacy effects of climate, evident in semi-arid lands (Hawkes et al., 2017), also appear to impact microbial enzyme activity and its response to substrate C and N availability in boreal forest soils, though temperature sensitivity of biomass-specific $CO_2$ release does not appear to change across long timescales of exposure to warming (Min et al., 2019).

Susan Ziegler 2021-4-7 11:52 AM

Susan Ziegler 2021-3-10 11:45 AM
**Moved (insertion) [1]**

Susan Ziegler 2021-3-13 3:22 PM

cold forests
Lower N, higher carbohydrate & bioreactivity

Observed respiratory responses

Predicted changes in SOM and R-CO$_2$

$Q_{10}= 3.6^*$    $Q_{10}= 2.3^*$

$>\delta^{13}C\ CO_2$    $<\delta^{13}C\ CO_2$

Increased A:OA & **Decreased C:N and %N** as THAA

Less change in A:OA, C:N and %N as THAA

warm forests
Higher N, lower carbohydrate & bioreactivity

$Q_{10}= 4.3^*$    $Q_{10}= 2.4^*$

$>>\delta^{13}C\ CO_2$    $<\delta^{13}C\ CO_2$

**Increased A:OA** & Decreased C:N and %N as THAA

Less change in A:OA, C:N and %N as THAA

*Laganiere et al. (2015); Podrebarac et al. (2016)

[revised manuscript text omitted]

between

**Page 2: [4] Deleted**      **Susan Ziegler**      **2021–03–13 3:22 PM**

between

**Page 3: [5] Moved to page 2 (Move #1)**   **Susan Ziegler**      **2021–03–10 11:45 AM**

Differences between SOC and SON composition between climate regions are consistent with the differences in the bioreactivity of these soils (Laganière et al., 2015) and the temperature responses of respiration of the whole soil profiles (Podrebarac et al., 2016). The colder forest soils exhibit an elevated carbohydrate content (Ziegler et al., 2017) while soil organic N content and processing, assessed through $\delta^{15}N$ and total hydrolyzable amino acid content and composition, indicates greater availability and turnover of N within the warmer climate soils along this transect (Philben et al., 2016).

**Page 3: [6] Deleted**      **Susan Ziegler**      **2021–03–10 11:49 AM**

This suggests that the soil horizon connectivity promoted by the whole profile enabled microbial access to substrates or enhanced substrate use that promoted the observed temperature responses not observed in the isolated soil horizons.

**Page 3: [7] Deleted**      **Susan Ziegler**      **2021–03–13 3:26 PM**

The prevalence and relevance of cross-horizon substrate use is unknown.

**Page 4: [8] Deleted**      **Susan Ziegler**      **2021–03–13 3:36 PM**

 relatively

**Page 4: [8] Deleted**      **Susan Ziegler**      **2021–03–13 3:36 PM**

 relatively

**Page 4: [8] Deleted**      **Susan Ziegler**      **2021–03–13 3:36 PM**

 relatively

**Page 4: [8] Deleted**      **Susan Ziegler**      **2021–03–13 3:36 PM**

 relatively

**Page 4: [8] Deleted**      **Susan Ziegler**      **2021–03–13 3:36 PM**

 relatively

**Page 4: [9] Deleted**      **Susan Ziegler**      **2021–03–13 3:44 PM**

m

**Page 4: [9] Deleted**      **Susan Ziegler**      **2021–03–13 3:44 PM**

m

**Page 4: [10] Deleted**      **Susan Ziegler**      **2021–04–07 12:10 PM**

When horizons are separated, the exchange and availability of labile C substrates across horizons is inhibited, potentially altering microbial decomposition processes and their response to temperature.

The relative availability of N can also greatly impact the strategies of microbial communities, their response to temperature and resulting rates of respiration (Billings and Ballantyne, 2013). Availability of soil N and its C:N ratio changes with soil depth, thereby representing another feature potentially responsible for differences in the microbial respiratory response to temperature between whole soil profiles and the sum of the same soils incubated in isolation.

| Page 4: [10] Deleted | Susan Ziegler | 2021–04–07 12:10 PM |
|---|---|---|

When horizons are separated, the exchange and availability of labile C substrates across horizons is inhibited, potentially altering microbial decomposition processes and their response to temperature. The relative availability of N can also greatly impact the strategies of microbial communities, their response to temperature and resulting rates of respiration (Billings and Ballantyne, 2013). Availability of soil N and its C:N ratio changes with soil depth, thereby representing another feature potentially responsible for differences in the microbial respiratory response to temperature between whole soil profiles and the sum of the same soils incubated in isolation.

| Page 4: [11] Deleted | Susan Ziegler | 2021–03–13 3:40 PM |
|---|---|---|

W

| Page 4: [11] Deleted | Susan Ziegler | 2021–03–13 3:40 PM |
|---|---|---|

W

| Page 4: [12] Moved to page 4 (Move #4) | Susan Ziegler | 2021–03–13 3:44 PM |
|---|---|---|

more fungal dominated communities in surface horizons may access N-rich, higher $E_a$ substrates, from deeper soil horizons a mechanism found to support priming effects in some soils (Li et al., 2017).

| Page 4: [13] Deleted | Susan Ziegler | 2021–04–07 12:14 PM |
|---|---|---|

those

| Page 4: [13] Deleted | Susan Ziegler | 2021–04–07 12:14 PM |
|---|---|---|

those

| Page 4: [13] Deleted | Susan Ziegler | 2021–04–07 12:14 PM |
|---|---|---|

those

| Page 4: [13] Deleted | Susan Ziegler | 2021–04–07 12:14 PM |
|---|---|---|

those

| Page 4: [13] Deleted | Susan Ziegler | 2021–04–07 12:14 PM |
|---|---|---|

those

| Page 4: [13] Deleted | Susan Ziegler | 2021–04–07 12:14 PM |
|---|---|---|

those

[revised manuscript text omitted]

---

## Author Response (AR2)

Dear Editor and reviewers.

Thank you for the thorough review and constructive criticism on our paper. The whole process has greatly improved the manuscript. We have addressed the reviewers' comments completely in this revision. This includes edits to both Figures 1 and 2 including the caption to Figure 1 to better convey the hypotheses depicted there in the right side of the figure. We have also edited the introduction and added some mechanistic explanation within discussion section 4.2 in addition to other edits as pointed out by both reviewers.

Point by point response to each reviewer.

Reviewer 1

I find that my previous comments have been well addressed. I have only a few minor remaining comments (see below).

Technical corrections
Line 20: "comes"
    Response: Corrected.

Line 24: Should this be "each other"?
    Response: Corrected.

Line 54: Explain the abbreviation "Ea".
    Response: Corrected.

Line 61: "elevated" and "enhanced": Compared to what?
    Response: Corrected.

Figure 2: Is it on purpose that the "mesocosms built" field for H is empty?
    Response: This was also raised by the other reviewer. We have carefully reviewed Figure 2 to see what the issue was and could not find anything missing from the H horizon information provided in the figure. We did, however, better align the middle column providing each separate horizon and corrected a couple of inconsistences which we hope addresses this issue which may have been from the formatting into the final document.

Line 269: "resulted"
    Response: Corrected.

Line 294: "whole"
    Response: Corrected.

Reviewer 2

I think the author´s have properly addressed my comments. Still there are a couple of issues that needs consideration:
    Response: All comments provided in the attached revision were addressed and can now be found in tracked changes in the new revision provided.

I am not sure that hypotheses are well explained in Figure 1. The representation of the hypotheses in the right hand illustrations is a bit confuse, and texts within the boxes not clear. A better, simpler, representation and explanation in the caption should be possible

Response: Understood, Figure 1 has now been edited in two ways. First the text has been reduced in that far right portion of the figure to more generalize the hypotheses.  Second, the figure caption has been edited to better clarify the hypotheses stated there as well. We hope this helps to better represent the hypotheses to the reviewer/reader.

On the other hand, reading the intro and hypotheses it is a bit unclear what is the mechanisms the author´s want to test in Hypotheses 2: Does they want to test whether fungi are using labile C from L to degrade N-rich substrate from F and H? or does the labile C from L leaches to F and H?

Response: Parts of the introduction have now been edited to better clarify the stated hypotheses.  The hypotheses include both possibilities – that is use of labile C leached from the L to degrade more slow-turnover substrates from the F and H but also the possibility of specifically enhancing fungal use of L substrates and their degradation of more N-rich substrates from the lower F and H horizons.

Interesting, thought, that N cycling is not altered by temperature or priming whatsoever. Which are the mechanisms that allows increasing rates of mineralization without the need for extra N use?? Perhaps a couple of sentences should be added where the authors explain what may be happening so that the increase in the use of C does not translate into increases in the use of N (section 4.2.)

Response: Good point here. A couple of sentences were added to section 4.2 to help clarify the possible mechanisms responsible for this observation.

Some editing will be needed in the intro section (see enclosed manuscript)
Response: All edits as suggested were made and tracked in this new revision.

Some more comments in the Manuscript enclosed
Response: All comments were addressed with the further revisions made in the manuscript.

---

## Author Response (AR4)

Response to Editor's Comments

I have given your revised manuscript another look, and think that it should become acceptable after some further revision of the text. Please give the manuscript a careful read and improve the phrasing / wording to increase readability and clarity as much as possible. To illustrate my point, I have highlighted some critical passages in the introduction and legend of Fig. 1 in the attached file.

Furthermore I ask you to add a concise conclusion section, which highlights the main insights of your study in relation to the hypotheses and ends with a broader perspective.

Thank you for your input on the readability and clarity of the writing.  This prompting has helped to clarify the text throughout after a thorough review of both the main manuscript and the supplemental materials.

The final discussion section 4.3 originally contained the main insights from the study as well as the broader implications and perspective.  Therefore, that section was edited to more clearly place those in relation to the hypotheses originally posed, and the section was renamed as the conclusion.